# ALTA: Compiler-Based Analysis of Transformers

**Peter Shaw**[1], **James Cohan**[2], **Jacob Eisenstein**[1], **Kenton Lee**[1], **Jonathan Berant**[1],
**Kristina Toutanova**[1]

[1] *Google DeepMind,* [2] *Google*

**Reviewed on OpenReview:** *https://openreview.net/forum?id=h7S1wl9xiR*

## Abstract

We propose a new programming language called ALTA and a compiler that can map ALTA programs to Transformer weights. ALTA is inspired by RASP, a language proposed by Weiss et al. (2021), and Tracr (Lindner et al., 2023), a compiler from RASP programs to Transformer weights. ALTA complements and extends this prior work, offering the ability to express loops and to compile programs to Universal Transformers, among other advantages. ALTA allows us to constructively show how Transformers can represent length-invariant algorithms for computing parity and addition, as well as a solution to the SCAN benchmark of compositional generalization tasks, without requiring intermediate scratchpad decoding steps. We also propose tools to analyze cases where the expressibility of an algorithm is established, but end-to-end training on a given training set fails to induce behavior consistent with the desired algorithm. To this end, we explore training from ALTA execution traces as a more fine-grained supervision signal. This enables additional experiments and theoretical analyses relating the learnability of various algorithms to data availability and modeling decisions, such as positional encodings. We make the ALTA framework — language specification, symbolic interpreter, and weight compiler — available to the community to enable further applications and insights.[1]

## 1 Introduction

There has been significant discussion and debate about the degree to which Transformers can perform compositional generalization and "System 2" reasoning, prompted by negative results on various evaluations for certain classes of Transformers (e.g., Dziri et al., 2023; Qiu et al., 2023; Shaw et al., 2021; Wu et al., 2024; Delétang et al., 2022; Mitchell et al., 2023; Valmeekam et al., 2023). Do such negative results reflect some mutable aspect of how such models were trained, or more fundamental architectural limitations? To better understand the conceptual limitations of Transformers, it would be useful to have an interpretable framework for understanding whether and how Transformers can represent and learn solutions to various tasks of interest. Such a framework could also potentially help elucidate a path towards improving these capabilities.

We present a new framework for compiling interpretable, symbolic programs to Transformer model weights. The framework is based on a new programming language called ALTA, A Language for Transformer Analysis. It includes an interpreter for symbolically executing ALTA programs, and a compiler for converting ALTA programs to Transformer model weights. ALTA is inspired by prior work that introduced a programming language for Transformers called RASP (Weiss et al., 2021), and prior work that built a compiler from RASP programs to model weights called Tracr (Lindner et al., 2023). ALTA complements and extends this prior work, with two key conceptual differences.

First, ALTA supports dynamic control flow operations such as loops. While Zhou et al. (2023b) showed how RASP programs can be executed within the context of an auto-regressive decoder to implement some forms

---

[1]Code is available at `https://github.com/google-deepmind/alta`.

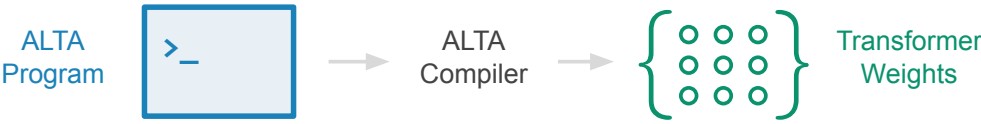

Figure 1: **Overview of ALTA.** We propose a new programming language called ALTA, and a "compiler" that can map ALTA programs to Transformer weights. ALTA is inspired by RASP, a language proposed by Weiss et al. (2021), and Tracr (Lindner et al., 2023), a compiler from RASP programs to Transformer weights. ALTA complements and extends this prior work, offering the ability to express loops and to compile programs to Universal Transformers, among other advantages.

of loops by leveraging scratchpads (Nye et al., 2021; Wei et al., 2022), ALTA can implicitly support such operations without relying on intermediate decoding steps. This is useful to study because such additional decoding steps can be computationally inefficient and are non-differentiable, typically necessitating additional supervision. ALTA accomplishes this by compiling to Transformers with layer-wise weight sharing. From one perspective, Transformers with weight sharing are simply a special case of standard Transformers. However, they have also been shown to have an inductive bias that improves performance on compositional tasks (Csordás et al., 2021; Ontanon et al., 2022; Yang et al., 2024), which warrants further study. ALTA also supports a conditional computation mechanism, enabling compilation of programs to Universal Transformers (Dehghani et al., 2019), thereby enabling new constructive expressivity results for this class of models.

Second, ALTA represents the computation of the MLP sub-layer as a sparse set of *transition rules*. We show that this enables compilation of complex programs to reasonably sized Transformers. In contrast, Tracr compiles functions over multiple variables expressed in RASP to dense lookup tables encoded in the MLP parameters, which can suffer from combinatorial explosion in the number of possible variable combinations. The ALTA compiler can leverage the sparsity expressed in the set of transition rules to reduce the number of MLP hidden dimensions required, in some cases by many orders of magnitude. Additionally, representing the MLP sub-layer computation as a set of sparse transition rules supports new insights into the generalization potential of MLP layers, and therefore of Transformers, which we explore both theoretically and empirically.

We highlight two primary applications of this framework. First, we show new constructive expressivity results for Transformers and Universal Transformers. This includes showing how Transformers can implement length-invariant algorithms for computing parity and addition. We also demonstrate a shift-reduce parsing algorithm that solves the SCAN (Lake & Baroni, 2018) benchmark of compositional generalization tasks. Second, we provide tools to analyze cases where the expressibility of an algorithm is established, but end-to-end training on a given training set fails to induce behavior consistent with the desired algorithm. Specifically, we propose to use intermediate supervision from ALTA execution traces over a given training set as a learning signal. In some cases, we show this additional supervision is sufficient to learn the desired algorithm, but in other cases failures can highlight limitations of the underlying architecture due to, e.g., the type of positional encoding used. To complement this empirical assessment, we also introduce the analytical notion of whether a program is *minimal* with respect to a training set, based on whether certain components of a program could be removed without affecting the training set predictions. We prove that if a program is not minimal with respect to a training set, then the compiled model will contain parameters that can be freely changed without affecting any predictions on the training set, i.e. some parameters are under-specified by the training set, even with intermediate supervision. We demonstrate cases where this analysis predicts that test set performance would be under-specified by a given training set, and show agreement with empirical results from training with intermediate supervision. We hope these tools can help provide insights to bridge the gap between expressibility and learnability of algorithms in Transformers.

In summary, our main contributions are as follows:

- We propose a new programming language called ALTA, and a compiler that can map ALTA programs to Transformer weights. ALTA supports loops and the ability to compile programs to Universal Transformers, among other advantages. (Section 2)

- We use ALTA to demonstrate new constructive expressivity results for Universal Transformers, showing how they can express length invariant algorithms for computing parity and addition, as well as a solution to SCAN. (Sections 3-4)

- We show how the ALTA framework can be used to analyze the learnability of various algorithms with respect to a training set, both theoretically and empirically via trace supervision. (Sections 3-4)

- We make the ALTA framework available to the community to support further applications and insights.

## 2 Proposed Framework

Here we give an overview of how ALTA programs are specified, their computational model, and how they can be compiled to Transformers. More details on the ALTA program API is in Appendix A.1, and compilation details and examples are in Appendix A.2.

### 2.1 Overview

We give an example of an ALTA program in Figure 2. An ALTA program specification includes three key components: a set of variables, a set of attention heads, and a "MLP function". We explain each of these below in the context of how they affect the execution of an ALTA program. The computational model of an ALTA program aligns closely with the computational model of a Transformer (Vaswani et al., 2017). In this paper we focus on *encoder-only* Transformers for simplicity. However, we note that ALTA programs can alternatively be executed in the context of a *decoder-only* Transformer. This involves adding a causal attention mask and outer auto-regressive decoding loop, but does not otherwise affect the definition and compilation of ALTA programs (see Appendix A.3). We also focus on Transformers with layer-wise weight sharing, i.e. where all attention and MLP parameters are shared across layers. We also support Universal Transformers which have an input-dependent number of layers.

Notably, not all types of computation expressible by Transformers can be represented in ALTA, i.e., the range of the ALTA compiler is a relatively small subspace of all possible parameter values. For example, ALTA has limited support for numeric computation and does not support modeling of probabilistic output distributions. However, ALTA provides broad support for implementing various types of deterministic algorithms.

The ALTA framework includes an *interpreter*, which symbolically executes a program, and a *compiler* which compiles programs to Transformer weights. The input to an ALTA program $P \in \mathcal{P}$ is a sequence of integers inputs ($\in \mathcal{X}$) and the output is a sequence of integers of equal length ($\in \mathcal{Y}$). The interpreter implements a function $I : \mathcal{P} \times \mathcal{X} \to \mathcal{Y}$. The interpreter is useful for development and understanding the computational model in an abstract way. The compiler implements a function $C$ such that $\theta = C(P)$ where $T(\mathbf{x}, \theta) \approx I(P, \mathbf{x})$ for all $\mathbf{x} \in \mathcal{X}$ and where $T$ denotes the output of a Transformer encoder. The equality holds up to the limits of numerical approximation for well formed programs.

### 2.2 Variables

Similarly to Lindner et al. (2023), we adopt the *residual stream* view of Transformers as proposed by Elhage et al. (2021). In this view, the attention and MLP sub-layers within the Transformer read and write to the residual stream, which is represented by the activations between these sub-layers. While Transformers represent the residual stream for each element as a vector, our interpreter represents the residual stream for each element symbolically as a mapping of variables to values. The residual stream of the interpreter for the

```python
vars = {
    # Initialize parity with input.
    "parity": var(range=2, input_init_fn=lambda x: x),
    # Whether parity has been updated.
    "done": var(range=2, position_init_fn=lambda x: x == 0),
    # Position of current element.
    "idx": var(range=NUM_POS, position_init_fn=lambda x: x),
    # Index of preceding element.
    "idx_left": var(range=NUM_POS, position_init_fn=lambda x: max(0, x - 1),
}

attention_heads = {
    # Values of 'parity' and 'done' for preceding element.
    "parity_left": qkv("idx_left", "idx", "parity")
    "done_left": qkv("idx_left", "idx", "done")
}

def ffn_fn(z):
  if not z["done"] and z["done_left"]:
    # Update parity based on parity of preceding element.
    z["parity"] = z["parity_left"] ^ z["parity"]
    z["done"] = 1

return program_spec(vars=vars, heads=attention_heads, ffn_fn=ffn_fn,
                    output="parity", halt_spec=halt_spec("done", 1),
                    input_range=2, position_range=NUM_POS)
```

Figure 2: **Example ALTA Program.** The parity program computes whether a given binary sequence contains an even or odd number of "1" tokens. For an input of length $N$, the `parity` variable of the final input element will equal the parity of the overall sequence after $N-1$ layers, and computation will halt. The program specification contains all of the necessary information to compile the program to a Transformer.

parity program of Figure 2 is shown in Figure 3. The set of variables and their possible values are specified by the program. There are three kinds of variables in ALTA: categorical variables have bounded integer values, numerical variables have real-valued values, and set variables have sets of bounded integer values. Variables representing the output of attention heads can also take on a *null* or *undefined* value (see §2.3). We establish a bijective mapping between variable assignments and activation vectors. Each possible value of a categorical variable is assigned a standard basis vector in the activation space, i.e., a one-hot encoding. Set variables are similarly represented, but with a multi-hot encoding. The scalar value of a numerical variable is directly represented in a single dimension. This mapping can be seen as establishing an approximate isomorphism with respect to the sub-layer operations of the interpreter and those of a compiled Transformer.

## 2.3   Execution

In this section, we explain the execution of an ALTA program in the interpreter, and summarize how each operation is encoded in a compiled Transformer, with more details in Appendix A. We denote the value of variable `foo` for element $i$ at sub-layer $k$ as $z^k_{\langle i, \text{foo} \rangle}$. Let $z^k_{\langle :, \text{foo} \rangle}$ denote the vector of values for `foo` across all elements at sub-layer $k$.

**Initialization**   Given an input sequence $\mathbf{x} = \langle x_1, x_2, \cdots, x_{|\mathbf{x}|} \rangle$, we initialize every variable at every position. Specifically, for a variable `foo`, we initialize $z^0_{\langle i, \text{foo} \rangle}$ as a function of $x_i$, a function of the positional index $i$, or as a constant, based on how the initialization for `foo` is specified in the program. This operation is encoded in the parameters of the Transformer's embedding tables for input and position values. The number of possible input values and possible positional indexes must be specified to the compiler. Alternatively, positional embeddings can be omitted if no variable is initialized as a function of position.

**Encoder Loop**   We then proceed to iteratively execute the self-attention sub-layer and the MLP sub-layer, which share parameters across all layers. A dynamic halting criterion, as proposed for the Univeral Transformer (Dehghani et al., 2019), can optionally be specified by the program, which consists of specify-

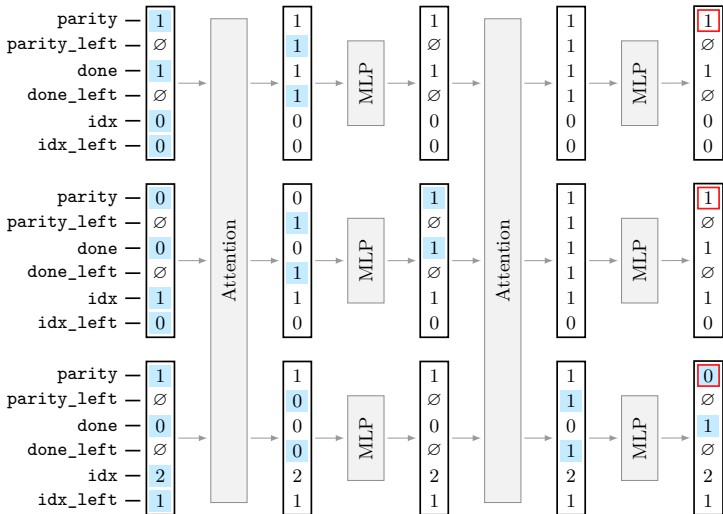

Figure 3: Visualization of the interpreter's symbolic residual stream for the parity program shown in Figure 2, for the input sequence $[1, 0, 1]$. The computed output sequence after the second layer is $[1, 1, 0]$, which corresponds to the parity of the input sequence up to the current position, with the final value containing the parity of the entire input sequence. Output values are outlined in red, and values that have changed are highlighted in blue.

ing which variable and corresponding value indicate that computation has completed for a given element. Alternatively, a maximum number of layers can be specified as an argument to the interpreter or when running a compiled Transformer. Similarly to Tracr (Lindner et al., 2023), our compiled Transformers do not include layer normalization operations, which simplifies compilation. Otherwise, the self-attention and MLP sub-layers align with those of a standard Transformer, and are described below.

**Self-Attention** For each attention head, the interpreter computes a selection matrix, containing a "weight" for every pair of inputs, and uses this to aggregate information across positions. A simplifying assumption of ALTA, similarly to RASP, is that this matrix is binary. Attention heads in ALTA are specified by pointers to query, key, value, and output variables. Each head must have a unique output variable. The query variable must be a categorical or set variable, the key variable must be categorical, and the value and output variables must both be either categorical or numerical. For each attention sub-layer $k$, every attention head updates the value of some output variable:

$$z_{\langle:,\text{out}\rangle}^{k+1} = \texttt{aggregate}(\texttt{select}(z_{\langle:,\text{query}\rangle}^{k}, z_{\langle:,\text{key}\rangle}^{k}), z_{\langle:,\text{value}\rangle}^{k}),$$

where `query`, `key`, `value`, and `out` are the variable names specified by the given attention head. The definition of `select` is similar to that used by RASP. The main difference is that we do not allow specifying a custom binary predicate as an argument to `select`, which simplifies compilation.[2] The `select` operation returns a square selection matrix, $S_{i,j}$, where $S_{i,j} = [\![z_{\langle i,\text{key}\rangle}^{k} = z_{\langle j,\text{query}\rangle}^{k}]\!]$ if `query` is categorical and $S_{i,j} = [\![z_{\langle i,\text{key}\rangle}^{k} \in z_{\langle j,\text{query}\rangle}^{k}]\!]$ if `query` is set-valued. The `aggregate` operation returns a new sequence of values $z_{\langle:,\text{out}\rangle}^{k+1}$. Each value $z_{\langle i,\text{out}\rangle}^{k+1}$ is determined by aggregating over the set of *selected values*, $\{z_{\langle j,\text{value}\rangle}^{k} | S_{i,j} = 1\}$ specified by the selection matrix row $S_{i,:}$. When `value` is numeric, `aggregate` is defined the same as in RASP, outputting the average of the selected values, and will be undefined if no value is selected. When `value` is categorical, the output will be undefined if there is not exactly one value selected. The interpreter will raise an exception if any undefined variable is used as input to any operation in subsequent layers, as the encoding of an undefined variable is not well specified in a compiled model.

---

[2]While this may seem to restrict the expressivity of the query, key, and value projections in the Transformer, we note that the MLP sub-layer prior to the attention sub-layer can set the query, key, and value variables to arbitrary values. By using set variables, it is still possible to specify arbitrary binary selection matrices.

These operations can be encoded in the parameters of the query, key, value, and output projections for a given attention head. A scalar hyperparameter controls the degree to which the softmax operation approximates generating a binary selection matrix. Each attention sub-layer is followed by a residual connection. We also support the option of using relative position representations (Shaw et al., 2018) by specifying a mask of relative positions that each attention head can attend to, which is applied to the selection matrix following the `select` operation. This binary mask is compiled to relative position biases using the parameterization of Raffel et al. (2020).

**MLP Function**  The MLP sub-layer implements a mapping from a set of variable assignments to a new set of variable assignments, which is applied at every element. For compilation and analysis purposes, ALTA programs internally represent this operation as a set of *transition rules*, which can be interpreted as logical implications with conjunctive antecedents. For example, here is one of the transition rules for the example program in Figure 2:

$$
\overbrace{z^{k+1}_{\langle\cdot,\texttt{parity}\rangle} = 1}^{consequent} \impliedby \overbrace{z^{k}_{\langle\cdot,\texttt{done}\rangle} = 0 \ \land \ z^{k}_{\langle\cdot,\texttt{done\_left}\rangle} = 1 \ \land \ z^{k}_{\langle\cdot,\texttt{parity\_left}\rangle} = 1 \ \land \ z^{k}_{\langle\cdot,\texttt{parity}\rangle} = 0}^{antecedent}
$$

When the *antecedent* of a rule is satisfied by the MLP input (i.e., all of the conditions hold with respect to the variable assignments at the MLP input), then the *consequent* determines the value of some *output variable* for the next sub-layer. By construction, for a given output variable, there should never be more than one rule satisfied by the MLP input. If no rule is satisfied, then the value of that output variable is unchanged from the MLP input. We also include transition rules that ensure that every attention output variable is set to a *null* value so that it can be updated by the next attention sub-layer without conflicting with the residual connection. No attention output variable can otherwise be the output variable of any transition rule.

The set of transition rules can be specified in two ways when defining ALTA programs. First, as shown in Figure 2, one can simply write a Python function with the signature shown. The set of transition rules can then be determined by executing this function for every possible set of variable assignments. In cases where this is not feasible, or where it is desirable to have more control over the set of transition rules, we offer an alternative API for specifying the set of transition rules more directly (see §A.1). In either case, the representation of numerical variable values in the antecedent of a transition rule is based on the set of discrete buckets specified for the given variable. Leveraging the sparsity represented in a set transition rules rather than compiling a lookup table consisting of all variable combinations can significantly reduce the number of MLP dimensions required.[3]

The set of transition rules is represented in the parameters of the MLP layers. We generate a 4-layer MLP with clipped ReLU activations. The first 2 layers are only responsible for converting numerical and set variables into a one-hot representation, representing the possible values of these variables. For numerical variables, these correspond to a specified set of discrete buckets. Note that if a program contains only categorical variables, these 2 layers could be omitted. The final 2 layers of the MLP are based on the set of transition rules. The parameters of these layers are compiled such that the hidden activations are a binary vector where each value corresponds to whether a particular transition rule was satisfied by the MLP input. Each row of the first matrix is a function of the antecedent of a particular rule, and each column of the second matrix is a function of the consequent of a particular rule. Each MLP sub-layer is followed by a residual connection.

**Output**  Each ALTA program specifies an output variable, which must be categorical. If execution terminates after $k$ sub-layers, and the output variable is `output`, then the program returns $z^{k}_{\langle:,\texttt{output}\rangle}$. Selecting the subset of dimensions associated with the output variable is encoded in the parameters of the output projection. The Transformer then computes a softmax over this one-hot vector, and outputs the argmax.

---

[3]For example, consider a string $x$ of length $N$ represented by a set of categorical variables, $x_1, x_2, \cdots, x_N$, each with $K$ possible values. We want to determine if $x$ is in some vocabulary consisting of $V$ strings. A naive lookup table approach requires $K^N$ hidden dimensions, but this function can be represented with only $V$ transition rules.

# 3 Expressibility and Learnability

While there are many potential applications for ALTA, we focus on two applications in this paper: new constructive expressivity demonstrations, and analysis of whether such algorithms are learnable given a particular training set, with varying amounts of supervision. We give an overview of these applications and our proposed analytical tools here, with results in Section 4.

## 3.1 Expressibility

There has been considerable interest in establishing the theoretical expressivity of various classes of Transformers (Pérez et al., 2021; Chiang et al., 2023; Yun et al., 2020; Feng et al., 2023; Merrill & Sabharwal, 2024). However, such theoretical works often do not demonstrate specific constructions of how Transformers can express algorithms of interest with limited resources, which can require manually specifying weight matrices. Tools like RASP, Tracr, and ALTA make it easier to construct such constructive demonstrations. For example, RASP and its extensions have supported many new expressivity results for Transformers (Zhou et al., 2023b; Angluin et al., 2023; Kazemnejad et al., 2024; Yang & Chiang, 2024; Strobl et al., 2024; Friedman et al., 2024).

Much of the recent work on the expressivity of Transformers has focused on Transformer decoders that can execute an input-dependent number of intermediate decoding steps before producing a final output (Pérez et al., 2021; Feng et al., 2023; Merrill & Sabharwal, 2024; Zhou et al., 2023b). While such intermediate decoding steps have been empirically successful at improving sequential reasoning capabilities (Nye et al., 2021; Wei et al., 2022), this typically requires additional supervision during training (Pfau et al., 2024). Additionally, leveraging such intermediate decoding steps may not be the most computationally efficient way to represent certain algorithms. Therefore, it is interesting to study whether and how Transformers can represent various algorithms *without* relying on such intermediate decoding steps, as an alternative or complementary approach. We use ALTA to provide new constructive expressibility results for Universal Transformer encoders for various resource bounds, detailed in Section 4. When considering a finite limit on the number of layers, such results also hold as a special case of standard Transformer encoders.

## 3.2 Learnability

In many cases, we can establish that an algorithm is expressible by a given class of Transformers, but training a model from this class on input and output examples of a particular algorithm can fail to induce a model that generalizes outside of the training set. It can be difficult to diagnose the reason for such failures, and to determine what to change regarding the architecture, training objective, or training data to improve generalization. We provide two tools to help bridge the gap between expressibility and learnability, which we discuss next.

**Trace Supervision**  We propose to use intermediate supervision from ALTA execution traces over a given training set as a learning signal. Given a program $P$ and a set of model inputs $\mathcal{X}$, we run $P$ for every input in $\mathcal{X}$, and extract the variable assignments at the input and output of every sub-layer, for every position. These traces can be used to derive *trace supervision*, which encourages the behavior of the model to align with that of the program being used to provide supervision. In our experiments, for simplicity, we focus on training the MLP parameters to reconstruct the desired output vector for each input vector, and compile the remaining parameters. In some cases, we show this additional supervision is sufficient to learn the desired algorithm, but in other cases failures can highlight limitations of the underlying architecture due to, e.g., the type of positional encoding used. We report results on the parity task in §4, and details of the training procedure in Appendix C.1.

**Theoretical Analysis**  To complement the empirical results from trace supervision, we also take a step towards analytically characterizing the conditions under which any particular ALTA program can be learned from a given training set. To this end, we introduce criteria for determining whether a program is *minimal* with respect to a training set, which depends on whether certain components of a program could be removed without affecting the training set predictions.

Here we give an overview of our theoretical analysis, which is detailed in Appendix B. We focus on the set of MLP parameters, and consider a setting similar to that of training with trace supervision, where we derive a set of MLP inputs, $\mathcal{D}$, corresponding to running a given program over some set of model inputs. The MLP parameters are specified by the set of transition rules, $\mathcal{R}$, in the given program. A rule set is *minimal* with respect to a set of MLP inputs, $\mathcal{D}$, if it is not possible to remove any rule or any constraint from any rule without changing the program output on some input in $\mathcal{D}$. Similarly to training with trace supervison, we consider a reconstruction loss over $\mathcal{D}$ that quantifies how well the MLP outputs correspond to the outputs specified by $\mathcal{R}$. Our results are with respect to a training objective that combines this reconstruction loss with a regularizer on the parameter weights. We show that:

**Theorem 1** (Informal)**.** *If the rule set $\mathcal{R}$ is* not minimal *with respect to $\mathcal{D}$, then the compiled MLP parameters are* not *a strict coordinate-wise local optimum of the regularized reconstruction loss.*

**Theorem 2** (Informal)**.** *If the rule set $\mathcal{R}$ is* minimal *with respect to $\mathcal{D}$, then the compiled MLP parameters are a strict coordinate-wise local optimum of the regularized reconstruction loss.*

Therefore, when the minimality conditions are met, we can conclude that there exists a local optimum point with respect to the training objective that corresponds to the behavior of the given program, although we are not guaranteed to find this point during training. See Theorems 1 and 2 in Appendix B for the formal statements and proofs.

We can apply this theory to analyze a particular program, training set, and test set. We can determine a *minimal version* of the program with respect to the training set by removing aspects of the program that do not change any predictions on the training set (Appendix C.2 describes the exact procedure used). We can then assess whether this minimal version *generalizes*, i.e., has the same behavior as the original program on a test set. This analytical test does not require training models or selecting hyperparameters, and provides an interpretable assessment of the potentially underspecified aspects of a program. Our theoretical analysis suggests that a Transformer trained with trace supervision from a program on a training set is more likely to generalize if the minimal version of the program generalizes, and we evaluate this in Section 4.

Notably, this notion of minimality can also help generalize some of the ideas introduced by Zhou et al. (2023b) with respect to RASP. As Zhou et al. (2023b) were interested in understanding length generalization, they proposed some restrictions on RASP programs that would otherwise be "difficult to learn". First, they proposed restrictions on programs containing certain operations over positional indices. Second, they excluded certain programs from consideration on an intuitive basis, such as a program for solving the parity task using sum and modulo operations. In both cases, such programs would not be minimal with respect to some finite length training set, according to our proposed criteria, as we show in Section 4.

## 4 Experiments and Analysis

We detail experiments and analysis on several tasks, with further details and results in Appendix D.

### 4.1 Parity

The parity task requires the model to compute whether a binary sequence contains an even or odd number of ones. The ability of transformers to learn parity has been studied extensively (Hahn, 2020), particularly the degree to which they exhibit length generalization (Bhattamishra et al., 2020; Chiang & Cholak, 2022; Ruoss et al., 2023; Delétang et al., 2022). Empirically successful solutions have relied on scratchpads (Anil et al., 2022; Zhou et al., 2022). Zhou et al. (2023b) used a variant of RASP to investigate why Transformers struggle with length generalization on parity and why scratchpads help.

We study three ALTA programs for computing parity detailed in Appendix D.1. First, the *Sequential (Absolute)* program computes parity by iterating through each position (one per layer), flipping a parity bit every time a one is encountered. While this program uses absolute positions to enable propagating information to neighboring tokens, we also consider a *Sequential (Relative)* version that uses relative positions instead. Finally, the *Sum + Modulo* program computes parity in a single layer, using an attention head to compute the total number of ones, and then computing a mod 2 operation in the following MLP sub-layer.

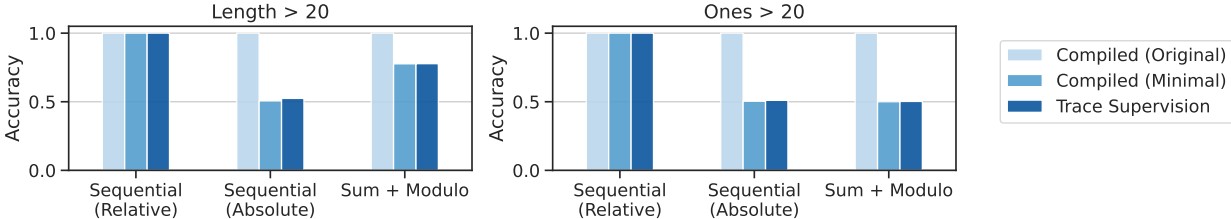

Figure 4: Generalization by length and number of ones for different parity programs in various settings. The training set consists of inputs of up to length 20 (and therefore up to 20 "ones") while the test set consists of inputs with lengths 21 to 40. While the original compiled models have perfect accuracy, the minimal versions of the programs exhibit different generalization behaviors and closely mirror the behaviors of the trace supervision experiments.

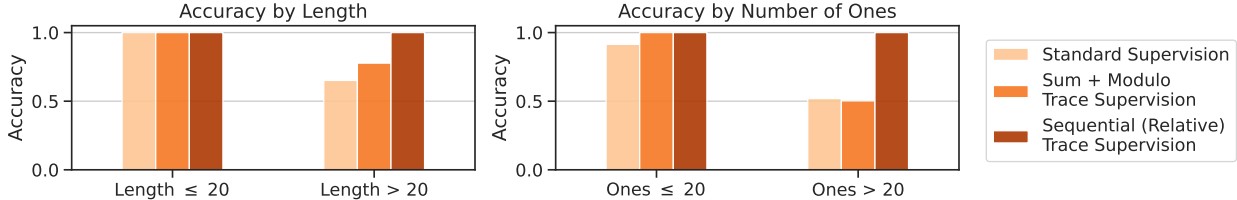

Figure 5: Accuracy by length and number of ones for Transformers trained with trace supervision vs. standard, end-to-end supervision. Transformers trained with standard supervision exhibit behavior similar to those trained with trace supervision from the *Sum + Modulo* program, i.e., they exhibit no generalization to numbers of ones greater than those seen during training, but they do exhibit some length generalization.

For each program, Figure 4 compares the generalization behavior of Transformers compiled from the original version, compiled from the minimal version, and trained from trace supervision. The standard compiled models achieve 100% accuracy, exactly emulating the behavior of the symbolic program, as expected. The minimal versions of these programs exhibit different generalization behavior. The minimal *Sequential (Absolute)* program does not generalize to examples longer than those seen during training, because the embeddings for positions not seen during training are not specified. The minimal *Sum + Modulo* program does not generalize to examples with more ones than those seen during training, as it does not contain transition rules related to the numerical values corresponding to larger numbers of ones than those seen during training. However, it can handle examples longer than the training examples if the numbers of ones were seen during training. Only the minimal *Sequential (Relative)* program generalizes to all input lengths. Notably, the trace supervision results exhibit the same generalization behavior as the minimal programs, as predicted in Section 3. The *Sequential (Relative)* program is also notable because it provides a constructive demonstration of how a Universal Transformer encoder can express a length-invariant solution to parity with a finite set of parameters, without relying on intermediate decoding steps.[4]

We also evaluated several different Transformer variants using standard, end-to-end supervision, with results shown in Figure 5. Theoretically, a transformer trained with weight sharing and at least as many layers as the longest examples in the train set could learn the minimal *Sequential (Relative)* algorithm and generalize to examples of length up to the number of layers. However, in practice, all variants exhibit behavior similar to that of the minimal *Sum + Modulo* program, i.e., they exhibit some degree of length generalization but do not generalize to examples with more ones than were seen during training. See §D.1 for additional results and experiment details.

---

[4]While Chiang & Cholak (2022) previously demonstrated how a Transformer can express a solution to parity for arbitrary lengths, their approach requires encoding activations and parameters with a degree of numerical precision that scales with the maximum input length. The *Sequential (Relative)* program does not have this limitation, but does require more layers of computation. Zhou et al. (2023b) also provide a length-invariant construction, but their approach requires intermediate decoding steps. See Appendix D.1 for details.

**Simplicity Bias** Prior work has attempted to understand the implicit inductive bias of Transformers in relation to a simplicity bias given some measure of complexity (Zhou et al., 2023b; Abbe et al., 2023; Bhattamishra et al., 2023; Tsoy & Konstantinov, 2024). For instance, inspired by the Minimum Description Length (MDL) principle (Rissanen, 1978; Grunwald, 2004), Zhou et al. (2023b) hypothesized that Transformers are biased towards learning behavior corresponding to the simplest RASP program that fits the training set, if one exists. Properties of the set of transition rules in an ALTA program can potentially provide new measures of Transformer complexity. For example, we can empirically compare the degree to which Transformers with layerwise weight sharing have an implicit bias towards learning "simpler" programs according to the number of transition rules. While the minimal *Sequential (Relative)* program contains 7 rules and the minimal *Sum + Modulo* program contains 31 rules, our results show that end-to-end model behavior is more consistent with the *Sum + Modulo* program. This indicates that Transformers, in this context, do not have an inherent simplicity bias that aligns with the number of transition rules expressed in ALTA. In this particular context, the end-to-end behavior is more consistent with the algorithm that requires the fewest layers to execute. The results for SCAN in Section 4.4 show a similar finding. This suggests a potentially interesting direction to explore for future work.

## 4.2 Addition

Another common benchmark for evaluating Transformer generalization is multi-digit addition. This task has been studied extensively (Nogueira et al., 2021; Liu et al., 2022), particularly with respect to length generalization (Zhou et al., 2024; Shen et al., 2023; Zhou et al., 2023b; Lee et al., 2024; Kazemnejad et al., 2024; Ruoss et al., 2023). While new positional encodings such as FIRE (Li et al., 2024) can improve performance, Transformers still struggle with length generalization on this task, unless provided with carefully constructed supervision over intermediate decoding steps (Zhou et al., 2023b; 2024).

In Appendix D.2 we detail an ALTA program with dynamic halting that can add two positive integers of unbounded size. This program compiles to a Universal Transformer encoder with relative position representations (as mentioned in Section 2, we use the parameterization of Raffel et al. (2020)). The number of layers required to compute the sum is $N+2$, where $N$ is the number of digits in the larger of the two inputs. Notably, the minimal version of this program with respect to a training set that includes only inputs with $\leq 3$ digits can generalize to unbounded input lengths.

We verified the correctness of our compiled model by considering samples of pairs of integer inputs with 1-50 digits. The compiled model achieved 100% accuracy across all instances, emulating the behavior of the symbolic program.

## 4.3 SUBLEQ

SUBLEQ is a single instruction language that has been shown to be Turing-complete when given access to infinite memory (Mavaddat & Parhami, 1988). Giannou et al. (2023) previously showed how a Looped Transformer can implement an interpreter for a variant of SUBLEQ. In Appendix D.3 we demonstrate an ALTA program for implementing an interpreter for a less restrictive version of SUBLEQ in a Universal Transformer encoder.

## 4.4 SCAN

The SCAN (Lake & Baroni, 2018) suite of compositional generalization tasks requires mapping natural language commands (e.g., "*jump twice*") to action sequences (e.g., `JUMP JUMP`). Certain train and test splits have been shown to be challenging for Transformer-based models (Keysers et al., 2020; Furrer et al., 2020; Qiu et al., 2022b; Kazemnejad et al., 2024). Empirically successful solutions have involved symbolic decompositions of some form (Shaw et al., 2021; Chen et al., 2020; Herzig & Berant, 2021; Qiu et al., 2022a; Zhou et al., 2023a).

In Appendix D.4, we demonstrate an ALTA program that solves the SCAN task. The program achieves 100% accuracy across all examples in the SCAN dataset. We also verified that the compiled model emulates the behavior of the program, similarly achieving 100% accuracy. First, the program executes a shift-reduce

parse of the input sequence, representing the parse as a tree. Second, the ALTA program decodes the output sequence by traversing the parse tree. The program represents the necessary variable-length data structures (a stack, parse tree, and buffer) using a variable number of input tokens. Compiled models require fewer than $2,000$ MLP hidden dimensions despite there being more than $10^{60}$ possible variable combinations in the program. This highlights the importance of sparsity, which ALTA enables by representing the MLP computation as a set of transition rules.

Notably, the *minimal version* of our program with respect to the training set generalizes to the test set, achieving 100% accuracy for all of the most challenging length-based and Maximum Compound Divergence (MCD) (Keysers et al., 2020) splits. Our ALTA program for SCAN thus gives a constructive demonstration of how Transformers can represent algorithms exhibiting systematic generalization: a finite set of transition rules and attention operations can be recombined in novel ways to process novel inputs.

Because our ALTA program decomposes the task into many small steps, it can require up to 512 layers to handle the longest SCAN examples. To understand whether Transformers trained with standard, end-to-end supervision can also benefit from such a large number of layers, we trained several Transformer variants, varying the total number of layers up to a maximum of 256 encoder layers and 256 decoder layers. However, all generalize poorly on the test splits, and increasing the number of layers does not improve generalization. Consistent with the parity results, end-to-end training does not learn a sequential algorithm that generalizes well, despite demonstrating that such an algorithm is expressible for a similar class of Transformers. See Appendix D.4 for additional results, experiment details, and discussion.

## 5 Discussion

**Limitations**  ALTA has many of the same limitations as RASP and Tracr with respect to the potential differences between compiled models and those learned in practice, as discussed by Lindner et al. (2023). In particular, the framework provides limited support for numerical computations and modeling probabilistic output distributions. The properties of compiled models may not reflect those of models learned in practice. However, ALTA can still be a useful tool if these limitations are kept in mind when interpreting results.

**Opportunities**  In this paper, we focused on analysis related to expressibility and learnability. However, there are many potential applications of the ALTA framework that could be interesting for future work. For example, we discuss more flexible alternatives for learning from trace supervision in Appendix C.1. ALTA could also potentially help develop test cases for interpretability tools. This was one of the primary motivations for Tracr, which has been applied to help design interpretability benchmarks (Thurnherr & Scheurer, 2024) and more interpretable Transformers (Friedman et al., 2023). Additionally, compiled components can potentially be integrated within learned models, such as circuits for arithmetic (Nanda et al., 2023) or induction heads (Akyürek et al., 2024). Finally, ALTA could potentially be extended to compare the relative expressivity of Transformers with self-attention compared with other architectures such as Linear Transformers (Katharopoulos et al., 2020) or State-Space Models (Gu et al., 2022). More broadly, we hope the ALTA framework can help the community better understand how Transformers can represent and learn various algorithms, and inspire new methods and insights.

### Acknowledgments

We would like to thank Anian Ruoss, David Lindner, Adam Fisch, Chris Dyer, and the anonymous reviewers for helpful feedback and discussions.

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

# A  Framework Details

In this section we provide additional details about the ALTA Framework, as introduced in §2. First, we detail the program API in §A.1, which is referenced by the ALTA programs in this paper. Second, we provide more details and examples of how programs are compiled to Transformer weights in §A.2.

## A.1  Program API Details

Here we detail the functions of the ALTA API used to build the programs shown in this paper. We refer the reader to our open-source implementation for further details.

**Variables and Attention Heads**   The module contains several methods for defining variable specifications:

- `var` defines a categorical variable, the most common variable type. The cardinality must be specified.

- `numerical_var` defines a numerical variable. A set of discrete buckets must be specified, and values are rounded to the closest bucket in the MLP sub-layer of compiled models.

- `set_var` defines a set variable. A set of sets of possible values must be specified.

For each of these functions, it is also necessary to specify how the variable is initialized.

There are two methods for defining attention heads:

- `qkv` defines an attention head, with arguments that specify the query, key, and value variables. Optionally, a set of relative positions can also be passed, as well as an explicit specification for the output variable.

- `relative_v` is a shorthand function for defining an attention head that attends to a specific relative position. The query and key are implicitly set to a single-valued categorical variable.

The output variable specification can be optionally specified explicitly, or is otherwise inferred from the type of the value variable. The output variable for each head is also included in the overall set of program variables.

**MLP Functions**   There are two ways to specify the MLP function, as mentioned in §2. For simplicity, the programs listed in this paper specify the MLP function as a Python function, which is passed a dictionary-like object for accessing and updating variable values. Alternatively, the set of transition rules can be specified directly, allowing more control to manage the scope of variables included in the antecedent of each rule, and avoid the combinatorial explosion of possible variable values for more complex programs. Figure 6 gives an example of specifying the transition rules for the parity program of Figure 2 using the `MLPBuilder` class. The class has two methods. The `get` method returns a generator over possible variable values. The class automatically tracks which variables and variable values are in scope. The `set` method generates a transition rule, generating the antecedent of the rule automatically based on the current scope. In almost all cases, this results in a more compact set of transition rules than specifying the MLP function as a Python function. All of the results in this paper related to analyzing the minimal versions of programs or computing the number of MLP hidden dimensions in compiled models are based on versions of programs where the transition rule set has been specified directly.

```
def get_transition_rules(variables, attention_heads):
  x = MLPBuilder(variables, attention_heads)
  for done in x.get("done"):
    if done != 1:
      for done_left in x.get("done_left"):
        if done_left == 1:
          x.set("done", 1)
          for parity_left, parity in x.get("parity_left", "parity"):
            x.set("parity", parity_left ^ parity)
  return x.rules
```

Figure 6: Function for directly specifying the set transition rules for the parity program of Figure 2.

## A.2  Compiler Details

In this section we will give an example of the compilation process introduced in §2, using the parity program of Figure 2 as an example.

**Notation**   Let $v_i^k$ denote the activation vector in a compiled model for element $i$ at sub-layer $k$.

**Encoding Variable Assignments**   Consider a set of variable assignments:

$$\texttt{parity} = 1$$
$$\texttt{parity\_left} = 0$$
$$\dots$$
$$\texttt{idx} = 2$$
$$\dots$$

For brevity, we only include a subset of program variables in this example. This set of assignments is represented as the following vector in a compiled model:

$$
\begin{bmatrix}
0 \\
1 \\
1 \\
0 \\
\dots \\
0 \\
0 \\
1 \\
\dots
\end{bmatrix}
\begin{matrix}
\texttt{parity} = 0 \\
\texttt{parity} = 1 \\
\texttt{parity\_left} = 0 \\
\texttt{parity\_left} = 1 \\
\\
\texttt{idx} = 0 \\
\texttt{idx} = 1 \\
\texttt{idx} = 2 \\
\end{matrix}
$$

**Initialization**   The input embedding is computed as: $z_i^0 = W_{x_i,:}^X + W_{i,:}^I$, where $x_i$ is the input token ID at position $i$, and $W^X$ and $W^I$ represent the embedding matrices for token and positional embeddings, respectively. For brevity, we only consider up to 3 positional indices for this example. We also omit some variables in the matrices below, and transpose them for clarity of the row and column labels. Note that

variables representing attention outputs, e.g. `parity_left`, are initialized to a *null* value.

$$
(W^X)^\top =
\begin{bmatrix}
1 & 0 \\
0 & 1 \\
0 & 0 \\
0 & 0 \\
\cdots & \cdots \\
0 & 0 \\
0 & 0 \\
0 & 0 \\
\cdots & \cdots
\end{bmatrix}
\begin{matrix}
\texttt{parity}=0 \\
\texttt{parity}=1 \\
\texttt{parity\_left}=0 \\
\texttt{parity\_left}=1 \\
\\
\texttt{idx}=0 \\
\texttt{idx}=1 \\
\texttt{idx}=2 \\
\end{matrix}
\qquad
(W^I)^\top =
\begin{bmatrix}
0 & 0 & 0 \\
0 & 0 & 0 \\
0 & 0 & 0 \\
0 & 0 & 0 \\
\cdots & \cdots & \cdots \\
1 & 0 & 0 \\
0 & 1 & 0 \\
0 & 0 & 1 \\
\cdots & \cdots & \cdots
\end{bmatrix}
\begin{matrix}
\texttt{parity}=0 \\
\texttt{parity}=1 \\
\texttt{parity\_left}=0 \\
\texttt{parity\_left}=1 \\
\\
\texttt{idx}=0 \\
\texttt{idx}=1 \\
\texttt{idx}=2 \\
\end{matrix}
$$

**Self-attention**  The self-attention operation sums over a set of attention heads:

$$
z_i^{k+1} = z_i^k + \sum_h o_i^h,
$$

where $o_i^h$ is the output of head $h$ parameterized by matrices $W_h^Q$, $W_h^K$, $W_h^V$, and $W_h^O$:

$$
o_i^h = W_h^O \sum_j \alpha_{ij} \cdot W^v z_j^k
$$

$$
\alpha_{ij} = \frac{e^{l_{ij}}}{\sum_k e^{l_{ik}}}
$$

$$
l_{ij} = (W_h^Q z_i^k)^\top W_h^K z_j^k.
$$

For simplicity, we ignore the constant scalar term commonly applied to the dot product. Simlarly to Tracr (Lindner et al., 2023), we also adopt the parameterization of Elhage et al. (2021) for the attention output, which can be shown to be equivalent to that of the original Transformer, but allows for a clearer exposition.

The parity program of Figure 2 has two attention heads. Here we describe the $W^Q$, $W^K$, $W^V$, and $W^O$ matrices corresponding to the attention head with query `idx_left`, key `idx`, value `parity`, and output `parity_left` (ommitted cells are zeros):

$$
(W^Q)^\top =
\begin{bmatrix}
\cdots & \cdots & \cdots \\
0 & 0 & 0 \\
0 & 0 & 0 \\
0 & 0 & 0 \\
\lambda & 0 & 0 \\
0 & \lambda & 0 \\
0 & 0 & \lambda
\end{bmatrix}
\begin{matrix}
\\
\texttt{idx}=0 \\
\texttt{idx}=1 \\
\texttt{idx}=2 \\
\texttt{idx\_left}=0 \\
\texttt{idx\_left}=1 \\
\texttt{idx\_left}=2
\end{matrix}
\qquad
(W^K)^\top =
\begin{bmatrix}
\cdots & \cdots & \cdots \\
\lambda & 0 & 0 \\
0 & \lambda & 0 \\
0 & 0 & \lambda \\
0 & 0 & 0 \\
0 & 0 & 0 \\
0 & 0 & 0
\end{bmatrix}
\begin{matrix}
\\
\texttt{idx}=0 \\
\texttt{idx}=1 \\
\texttt{idx}=2 \\
\texttt{idx\_left}=0 \\
\texttt{idx\_left}=1 \\
\texttt{idx\_left}=2
\end{matrix}
$$

$$
(W^V)^\top =
\begin{bmatrix}
1 & 0 \\
0 & 1 \\
0 & 0 \\
0 & 0 \\
\cdots & \cdots
\end{bmatrix}
\begin{matrix}
\texttt{parity}=0 \\
\texttt{parity}=1 \\
\texttt{parity\_left}=0 \\
\texttt{parity\_left}=1 \\
\end{matrix}
\qquad
W^O =
\begin{bmatrix}
0 & 0 \\
0 & 0 \\
1 & 0 \\
0 & 1 \\
\cdots & \cdots
\end{bmatrix}
\begin{matrix}
\texttt{parity}=0 \\
\texttt{parity}=1 \\
\texttt{parity\_left}=0 \\
\texttt{parity\_left}=1 \\
\end{matrix}
$$

where $\lambda$ is a hyperparameter that controls the degree to which the selection matrix approximates a binary-valued matrix, set to 100 by default.

**MLP Sub-layer** Our approach to compiling MLP parameters takes loose inspiration from Nielsen (2016), which provides a helpful visual exposition of how MLPs can encode arbitrary functions. The MLP function of the parity program can be represented with 7 transition rules, using the notation of Section 2:

$$R_1 : z^{k+1}_{\langle \cdot, \texttt{parity} \rangle} = 1 \impliedby z^k_{\langle \cdot, \texttt{done} \rangle} = 0 \ \wedge \ z^k_{\langle \cdot, \texttt{done\_left} \rangle} = 1 \ \wedge \ z^k_{\langle \cdot, \texttt{parity\_left} \rangle} = 1 \ \wedge \ z^k_{\langle \cdot, \texttt{parity} \rangle} = 0$$

$$R_2 : z^{k+1}_{\langle \cdot, \texttt{parity} \rangle} = 0 \impliedby z^k_{\langle \cdot, \texttt{done} \rangle} = 0 \ \wedge \ z^k_{\langle \cdot, \texttt{done\_left} \rangle} = 1 \ \wedge \ z^k_{\langle \cdot, \texttt{parity\_left} \rangle} = 1 \ \wedge \ z^k_{\langle \cdot, \texttt{parity} \rangle} = 1$$

$$R_3 : z^{k+1}_{\langle \cdot, \texttt{done} \rangle} = 1 \impliedby z^k_{\langle \cdot, \texttt{done} \rangle} = 0 \ \wedge \ z^k_{\langle \cdot, \texttt{done\_left} \rangle} = 1$$

$$R_4 : z^{k+1}_{\langle \cdot, \texttt{parity\_left} \rangle} = \varnothing \impliedby z^k_{\langle \cdot, \texttt{parity\_left} \rangle} = 0$$

$$R_5 : z^{k+1}_{\langle \cdot, \texttt{parity\_left} \rangle} = \varnothing \impliedby z^k_{\langle \cdot, \texttt{parity\_left} \rangle} = 1$$

$$R_6 : z^{k+1}_{\langle \cdot, \texttt{done\_left} \rangle} = \varnothing \impliedby z^k_{\langle \cdot, \texttt{done\_left} \rangle} = 0$$

$$R_7 : z^{k+1}_{\langle \cdot, \texttt{done\_left} \rangle} = \varnothing \impliedby z^k_{\langle \cdot, \texttt{done\_left} \rangle} = 1$$

Appendix B provides a more formal description of the MLP sub-layer and the compilation process. Here we provide an example of the compiled parameters $W^1$, $b^1$, and $W^2$ for the parity program (the second bias term, $b^2$, is always set to a vector of zeros). Note that we use a clipped ReLU function as the non-linearity. As described in Section A, there are also two initial MLP layers that are responsible for converting numerical and set variables into one-hot representations, but those are not necessary for the parity program we are considering which uses only categorical variables. We also omit the cells corresponding to `idx` and `idx_left` in the matrices and vector below (the corresponding cells are all zeros).

Each column of the transposed first matrix corresponds to a rule, with a 1 in cells corresponding to variable values in the *antecedent* of the rule. Each scalar in the bias vector has a value $1 - N$, where $N$ is the number of conjuncts in the antecedent of a given rule. This ensures that the output of the following non-linearity is 1 when all conjuncts are satisfied, and 0 otherwise. Each column of the second matrix is similarly associated with a rule, but the values are determined by the *consequent* of the rule. There is a 1 in the row corresponding to the new value of the output variable (or 0, if the output value is null), and a $-1$ corresponding to the value of the output variable in the antecedent of the rule.

$$(W^1)^\top = \begin{array}{c|ccccccc|l}
 & R_1 & R_2 & R_3 & R_4 & R_5 & R_6 & R_7 & \\
\hline
 & 1 & 0 & 0 & 0 & 0 & 0 & 0 & \texttt{parity} = 0 \\
 & 0 & 1 & 0 & 0 & 0 & 0 & 0 & \texttt{parity} = 1 \\
 & 0 & 0 & 0 & 1 & 0 & 0 & 0 & \texttt{parity\_left} = 0 \\
 & 1 & 1 & 0 & 0 & 1 & 0 & 0 & \texttt{parity\_left} = 1 \\
 & 1 & 1 & 1 & 0 & 0 & 0 & 0 & \texttt{done} = 0 \\
 & 0 & 0 & 0 & 0 & 0 & 0 & 0 & \texttt{done} = 1 \\
 & 0 & 0 & 0 & 0 & 0 & 1 & 0 & \texttt{done\_left} = 0 \\
 & 1 & 1 & 1 & 0 & 0 & 0 & 1 & \texttt{done\_left} = 1 \\
 & \cdots & \cdots & \cdots & \cdots & \cdots & \cdots & \cdots & \\
\end{array}$$

$$(b^1)^\top = \begin{array}{ccccccc}
R_1 & R_2 & R_3 & R_4 & R_5 & R_6 & R_7 \\
-3 & -3 & -1 & 0 & 0 & 0 & 0
\end{array}$$

$$W^2 = \begin{array}{c|ccccccc|l}
 & R_1 & R_2 & R_3 & R_4 & R_5 & R_6 & R_7 & \\
\hline
 & -1 & 1 & 0 & 0 & 0 & 0 & 0 & \texttt{parity} = 0 \\
 & 1 & -1 & 0 & 0 & 0 & 0 & 0 & \texttt{parity} = 1 \\
 & 0 & 0 & 0 & -1 & 0 & 0 & 0 & \texttt{parity\_left} = 0 \\
 & 0 & 0 & 0 & 0 & -1 & 0 & 0 & \texttt{parity\_left} = 1 \\
 & 0 & 0 & -1 & 0 & 0 & 0 & 0 & \texttt{done} = 0 \\
 & 0 & 0 & 1 & 0 & 0 & 0 & 0 & \texttt{done} = 1 \\
 & 0 & 0 & 0 & 0 & 0 & -1 & 0 & \texttt{done\_left} = 0 \\
 & 0 & 0 & 0 & 0 & 0 & 0 & -1 & \texttt{done\_left} = 1 \\
 & \cdots & \cdots & \cdots & \cdots & \cdots & \cdots & \cdots & \\
\end{array}$$

**MLP Example** Consider a set of variable assignments:

$$\texttt{parity} = 1$$
$$\texttt{parity\_left} = 1$$
$$\texttt{done} = 0$$
$$\texttt{done\_left} = 1$$
$$\dots$$

Let $z$ denote the vector encoding of these assignments:

$$z = \begin{bmatrix} 0 \\ 1 \\ 0 \\ 1 \\ 1 \\ 0 \\ 0 \\ 1 \\ \dots \end{bmatrix} \begin{matrix} \texttt{parity} = 0 \\ \texttt{parity} = 1 \\ \texttt{parity\_left} = 0 \\ \texttt{parity\_left} = 1 \\ \texttt{done} = 0 \\ \texttt{done} = 1 \\ \texttt{done\_left} = 0 \\ \texttt{done\_left} = 1 \end{matrix}$$

Here are the hidden activations in the MLP layer, which is binary vector corresponding to which rules are satisfied by the input:

$$h(z) = \sigma(W^1 z + b^1) = \begin{bmatrix} 0 \\ 1 \\ 1 \\ 0 \\ 1 \\ 0 \\ 1 \end{bmatrix} \begin{matrix} R_1 \\ R_2 \\ R_3 \\ R_4 \\ R_5 \\ R_6 \\ R_7 \end{matrix}$$

Note that $\sigma$ is a clipped ReLU non-linearity applied element-wise. Here is the output of the MLP sub-layer before and after the residual connection:

$$W^2 h(z) = \begin{bmatrix} 1 \\ -1 \\ 0 \\ -1 \\ -1 \\ 1 \\ 0 \\ -1 \\ \dots \end{bmatrix} \begin{matrix} \texttt{parity} = 0 \\ \texttt{parity} = 1 \\ \texttt{parity\_left} = 0 \\ \texttt{parity\_left} = 1 \\ \texttt{done} = 0 \\ \texttt{done} = 1 \\ \texttt{done\_left} = 0 \\ \texttt{done\_left} = 1 \end{matrix} \qquad W^2 h(z) + z = \begin{bmatrix} 1 \\ 0 \\ 0 \\ 0 \\ 0 \\ 1 \\ 0 \\ 0 \\ \dots \end{bmatrix} \begin{matrix} \texttt{parity} = 0 \\ \texttt{parity} = 1 \\ \texttt{parity\_left} = 0 \\ \texttt{parity\_left} = 1 \\ \texttt{done} = 0 \\ \texttt{done} = 1 \\ \texttt{done\_left} = 0 \\ \texttt{done\_left} = 1 \end{matrix}$$

Which corresponds to the expected output assignments:

$$\texttt{parity} = 0$$
$$\texttt{parity\_left} = \varnothing$$
$$\texttt{done} = 1$$
$$\texttt{done\_left} = \varnothing$$
$$\dots$$

where the attention output variables are set to *null* values so they are ready to be updated at the next self-attention layer.

### A.3 Decoder-only Extensions

While this paper focuses on using ALTA to compile programs for encoder-only Transformers, the framework can be extended to decoder-only models without significant modifications. Similarly, RASP was originally used to study encoder-only Transformers (Weiss et al., 2021), but was later extended to study decoder-only Transformers (Zhou et al., 2023b). The set of parameters necessary to specify an encoder-only and a decoder-only Transformer are the same, so the ALTA language specification and compiler do not require any changes. Only the interpreter and the Transformer implementation require minor modifications. First, it would be necessary to apply a causal attention mask. In the interpreter, this mask can be applied after determining the selection matrix. Second, rather than computing a single forward pass, the model should be run in the context of an auto-regressive decoding loop, i.e. where the previously decoded output is iteratively appended to the input sequence. For computational efficiency, caching can be used to avoid redundant computation for previously computed timesteps. We leave analysis of decoder-only Transformers to future work.

## B    Theoretical Analysis of Minimal Rule Sets and MLP Parameters

If a program is *minimal* with respect to some training set, can we draw any conclusions about whether the transformer implementation of that program will be *learned* on that same training set? We now explore this question from the perspective of the MLP parameters in the trace supervision setting. Specifically, **we identify conditions under which the compiled MLP parameters are a strict coordinate-wise local optimum of a regularized reconstruction loss.** Extensions to consider all parameters of the compiled transformer, end-to-end training objectives, or non-coordinate-wise local optima are left to future work.

### B.1    MLP Specification

Here we introduce notation that differs from that in other sections, but is more appropriate for the goals of this section. Let us focus on two components of an ALTA program $P$ for this analysis: a set of possible variable assignments, $\mathcal{V}$, and a set of rules, $\mathcal{R}$. The rules implicitly define an MLP function, $f_{\mathcal{R}} : \mathcal{V} \to \mathcal{V}$, which is a sub-component of the overall program. For simplicity, we consider only programs with categorical variables for this analysis, and exclude transition rules related to attention outputs.

**Variable Assignments**    An assignment $V \in \mathcal{V}$ is a tuple of $N_{\mathcal{V}}$ elements, with $V = \langle V_0, V_1, \cdots, V_{N_{\mathcal{V}}-1} \rangle$, where $V_i \in \{0, 1, \ldots, D_i - 1\}$ and $D_i$ is the dimensionality of variable $V_i$. The number and dimensionality of variables is given by the program specification.

**Transition Rules**    Let $\mathcal{R} = \langle R_1, R_2, \cdots, R_{N_R} \rangle$, where $R_i$ is a transition rule. As described informally in Section 2.3, transition rules consist of the following components:

- The *antecedent* of $R_i$, denoted $A_{R_i}$, consists of a set of $N_{R_i}$ *constraints* $(k, v)$ where each $k \in \{0, \cdots, N_{\mathcal{V}}-1\}$ refers to a *variable index*, and each $v \in \{0, \cdots, D_k - 1\}$ refers to a *variable value*. A transition rule is *satisfied* by an assignment $V \in \mathcal{V}$ if and only if $V_k = v$ for all $(k, v)$ in the rule's antecedent.

- The *consequent* of $k_{R_i}$ includes an the *output variable* index, denoted $k_{R_i}$, and a new value for this variable, denoted $v'_{R_i}$. Per construction, for every $R_i$ we require that the output variable appears in the antecedent of the rule. Let $v_{R_i}$ denote to the value associated with the output variable in the antecedent.

The rule $R_i$ can therefore be interpreted as updating the value of the variable $k_{R_i}$ from $v_{R_i}$ to $v'_{R_i}$ if $R_i$ is satisfied.

Finally, we also require that for any assignment $\in \mathcal{V}$, no more than one rule is satisfied for a given output variable. In other words, if two rules share the same *output variable* index, then they should never both be *satisfied* for any assignment $\in \mathcal{V}$.

**Logical MLP Function** We can define the logical MLP function $f_R : \mathcal{V} \to \mathcal{V}$ in terms of $\mathcal{R}$. Let $V' = f_R(V)$. Then:

$$V_k' = \begin{cases} v_{R_i}', & \text{if } \exists R_i \in \mathcal{R} \text{ that is satisfied by } V \\ V_k, & \text{otherwise.} \end{cases}$$

## B.2 Compiling MLP Parameters

As described in Section 2.3 and Appendix A.2, in our compiled models, we represent variable assignments as vectors, and compile the MLP function into the parameters of an MLP. Here we introduce notation and a more detailed description of the compilation procedure to support our theoretical analysis.

**Assignment Embeddings** In our compiled models, variable assignments are represented as $N_Z$-dimensional vectors. Let $\mathcal{Z} = \mathbb{R}^{N_Z}$ represent the space of such vectors, which include the inputs and outputs of MLP layers.

We can define a mapping $e_{\mathcal{V}} : \mathcal{V} \to \mathcal{Z}$. First, we define a bijective function $m(k, v)$ which for a given variable index $k \in \{0, 1, \dots, N_v - 1\}$ and corresponding variable value $v \in \{0, 1, \dots, D_k\}$, returns an index in $\{0, 1, \dots, N_Z - 1\}$. Let $\langle z_0, z_1, \cdots, z_{N_z - 1} \rangle = e_{\mathcal{V}}(V)$. Then:

$$z_i = [\![ \exists k, v \mid V_k = v \wedge m(k, v) = i ]\!]. \tag{1}$$

In other words, for a given assignment $V$, the vector $e_{\mathcal{V}}(V)$ will have a 1 for dimensions associated with variable values in $V$, and 0's elsewhere.

**MLP Function** The compiled MLP function $f_\theta : \mathcal{Z} \to \mathcal{Z}$ is defined by parameters $\theta \in \Theta$, where $\theta = \langle W^1, b^1, W^2 \rangle$, and $W^1 \in \mathbb{R}^{N_R \times N_Z}$, $b^1 \in \mathbb{R}^{N_R}$, and $W^2 \in \mathbb{R}^{N_Z \times N_R}$. The bias term in the second layer is set to all zeros in practice, and omitted here for simplicity.

For $z \in \mathcal{Z}$, we can the define the MLP function:

$$g(\theta, z) = W^1 z + b^1, \tag{2}$$
$$h(\theta, z) = \sigma(g(\theta, z)) \tag{3}$$
$$f(\theta, z) = W^2 h(\theta, z) + z \tag{4}$$

where $\sigma(z) = \max(0, \min(1, z))$ is a clipped ReLU activation function applied element-wise to the vector $z$.[5]

For notational convenience, let $f_i(\theta, z) := (f(\theta, z))_i$, i.e. denote the $i$th element of the output of $f$, and similarly for $g$, $h$. Following equations 4-2, the values of these individual scalars are:

$$g_i(\theta, z) = \sum_j W_{i,j}^1 z + b_i^1, \tag{5}$$

$$h_i(\theta, z) = \sigma(g_i(\theta, z)) \tag{6}$$

$$f_i(\theta, z) = z_i + \sum_j W_{i,j}^2 h_j(\theta, z) \tag{7}$$

---

[5]As noted by Pérez et al. (2021), the clipped ReLU function can be implemented in terms of the standard ReLU function, i.e. $\sigma(z) = max(0, z) - max(0, z - 1)$. Therefore, there is an equivalent formulation that uses additional layers but standard ReLU activations. However, we use clipped ReLU for simplicity.

**MLP Parameters**  We can define the compiled parameters $\hat{\theta} \in \Theta$ such that $e_{\mathcal{V}}(f_R(V)) = f_{\hat{\theta}}(e_{\mathcal{V}}(\mathbf{V}))$ for all $V \in \mathcal{V}$. Let $\hat{\theta} = \langle \hat{W}^1, \hat{b}^1, \hat{W}^2 \rangle$.

$$\hat{W}^1_{i,j} = \llbracket \exists\ (k,v) \in A_{R_i} \text{ s.t. } m(k,v) = j \rrbracket \tag{8}$$

$$\hat{b}^1_i = 1 - N_{R_i} \tag{9}$$

$$\hat{W}^2_{i,j} = \begin{cases} -1, & \text{if } m(k_{R_j}, v_{R_j}) = i \\ 1, & \text{if } m(k_{R_j}, v'_{R_j}) = i \\ 0, & \text{otherwise} \end{cases} \tag{10}$$

This construction ensures that for any assignment $V \in \mathcal{V}$, we have:

$$g_j(\hat{\theta}, e_{\mathcal{V}}(V)) = \begin{cases} 1, & R_j \text{ is satisfied by } V \\ 0, & N_{R_j} - 1 \text{ constraints of } R_j \text{ are satisfied by } V \\ -1, & N_{R_j} - 2 \text{ constraints of } R_j \text{ are satisfied by } V \\ \cdots, & \cdots \end{cases} \tag{11}$$

$$h_j(\hat{\theta}, e_{\mathcal{V}}(V)) = \llbracket R_j \text{ is satisfied by } V \rrbracket. \tag{12}$$

In other words, each element in the hidden layer activations corresponds to whether a particular rule was satisfied by the variable assignment represented in the input. Each column of $\hat{W}^2$ encodes an "update" to the assignments related to a particular rule. This construction also ensures that:

$$f(\hat{\theta}, e_{\mathcal{V}}(V)) = e_{\mathcal{V}}(f_{\mathcal{R}}(V)). \tag{13}$$

An example demonstrating equations 11, 12, and 13 is given in Appendix A.2.

## B.3  Minimal Rule Set

In Section 3 we introduced the notion of a *minimal program*. Here we focus on the minimality conditions relating only to the rule set and the corresponding MLP layer. We define a *minimal rule set*, encompassing the conditions necessary to obtain guarantees about a reconstruction loss landscape around the parameters of the compiled MLP.

**Definition 1.** *(minimal rule set) Given a dataset of $N$ variable assignments $\mathcal{D} = \langle V_1, V_2, \ldots, V_N \rangle$, a set of rules $\mathcal{R}$ is a **minimal rule set** over $\mathcal{D}$ if:*

- *It is not possible to remove any individual rule in $\mathcal{R}$ and not change the output of $f_{\mathcal{R}}$ for some example $V_n \in \mathcal{D}$.*

- *It is not possible to remove any individual constraint of the antecedent of any rule in $\mathcal{R}$ without changing the output of $f_{\mathcal{R}}$ for some example $V_n \in \mathcal{D}$.*

## B.4  Reconstruction Loss

We consider a setting related to that of training models with trace supervision, as discussed in §C.1. For program $P$ and set of model inputs $\mathcal{X}$, let $\mathcal{D} = \langle V_1, V_2, \ldots, V_N \rangle$ be the set of variable assignments at the input to the MLP for every position and layer when we run $P$ over $\mathcal{X}$. We use the shorthand:

$$z^n := e_{\mathcal{V}}(V_n), \tag{14}$$

to denote the vector encoding of assignment $V_n$.

Now let us define a *reconstruction loss*, $L_R$, over individual predictions. For model parameters $\theta$ and a given variable assignment $V_n$ with corresponding vector encoding $z^n = e_{\mathcal{V}}(V_n)$, the reconstruction loss quantifies how well the predicted variable encodings $f(\theta, z^n)$ match the variable encodings specified by $\mathcal{R}$, $e_{\mathcal{V}}(f_{\mathcal{R}}(V_n))$.

Per equation 13, this vector can alternatively be written in terms of compiled parameters $\hat{\theta}$ as $f(\hat{\theta}, z^n)$. For simplicity, we consider an $L_1$ loss over the predicted and expected encodings, which simplifies the analysis of the local minimum point. We write $L_R$ as:

$$L_R(\theta, z^n) = ||f(\theta, z^n) - f(\hat{\theta}, z^n)||_1. \tag{15}$$

Note that for all $n$:

$$L_R(\hat{\theta}, z^n) = 0. \tag{16}$$

### B.5 Coordinate-wise Local Minimum

As it is difficult to analyze whether a point is a strict local minimum, we will instead analyze whether a point is a coordinate-wise local minimum. Prior work has defined the notion of a coordinate-wise minimum, which is defined in terms of axis-aligned movements of parameter values (e.g., Tseng, 2001).

**Definition 2.** *(coordinate-wise local minimum) For a multivariate function $f : \mathcal{X}_1 \times \mathcal{X}_2 \times \ldots \times \mathcal{X}_n \to \mathbb{R}$, the value $\hat{X} = \langle \hat{x}_1, \hat{x}_2, \ldots, \hat{x}_n \rangle$ is a strict coordinate-wise local minimum iff for each $i$ $\exists \lambda > 0$, such that $f(\hat{x}_1, \ldots, \hat{x}_i, \ldots \hat{x}_n) < f(\hat{x}_1, \ldots, \hat{x}_i + \epsilon, \ldots \hat{x}_n)$ for all $\epsilon \in (-\lambda, \lambda)$.*

We will be evaluating a loss function at a semi-differentiable point (due to the non-linearity $\sigma$), so it is useful to note the following proposition:

**Proposition 1.** *Consider a multivariate function $f$ and point $\hat{X}$. If $f$ is a continuous function that is semi-differentiable at $\hat{X}$ with both left and right partial derivatives, and for every $i$ the right derivative with respect to $X_i$ evaluated at $\hat{X}$ is positive, $\frac{\partial^+ f}{\partial X_i}(\hat{X}) > 0$, and the left derivative with respect to $X_i$ evaluated at $\hat{X}$ is negative, $\frac{\partial^- f}{\partial X_i}(\hat{X}) < 0$, then $\hat{X}$ is a strict coordinate-wise local minimum as defined in Definition 2.*

This follows from the definition of left and right derivatives, and the requirement that $f$ is continuous.

### B.6 Main theorems

Suppose we learn $\theta$ by optimizing a regularized sum of reconstruction losses over a dataset,

$$L(\theta, \mathcal{D}, \alpha) = \alpha L_1(\theta) + \sum_n L_R(\theta, e_{\mathcal{V}}(V_n)) \tag{17}$$

$$= \alpha L_1(\theta) + \sum_n L_R(\theta, z^n),$$

with $\alpha > 0$ penalizing the sum of $L_1$ norms $||W^2||_1 + ||W^1||_1 + ||b^1||_1$. We consider the $L_1$ norm as the regularization term for simplicity.

We want to show that when the weights $\hat{\theta}$ are generated by compiling a set of rules that is *minimal* with respect to $\mathcal{D}$, then $\hat{\theta}$ is a coordinate-wise local optimum of $L$ for $\alpha < 1$.

**Theorem 1** (Formal). *If $\hat{\theta}$ is the compilation of a rule set $\mathcal{R}$ that is not minimal for $\mathcal{D}$, then $\hat{\theta}$ is not a strict coordinate-wise local optimum of $L(\theta, \mathcal{D}, \alpha)$.*

**Theorem 2** (Formal). *If $\hat{\theta}$ is the compilation of a rule set $\mathcal{R}$ that is minimal for $\mathcal{D}$, then $\hat{\theta}$ is a strict coordinate-wise local optimum of $L(\theta, \mathcal{D}, \alpha)$ for $\alpha < 1$.*

*Remark.* Both theorems can also be proved for a squared $L_2$ regularizer with coefficient $\alpha \in (0, 1/(2K))$, where $K = \max(1, \max_i N_{R_i} - 1)$ is the largest parameter magnitude in $\hat{\theta}$ and $2K$ is therefore the largest element of the gradient of the squared $L_2$ regularizer evaluated at $\hat{\theta}$.

### B.7 Proof of Theorem 1

*Proof.* There are two ways in which a ruleset might violate minimality with respect to $\mathcal{D}$.

- **We can remove rule $R_j$ without affecting the output of $f_{\mathcal{R}}$ on any example.** All rules affect the output when their antecedent conditions are met, so we can infer that the conditions of $R_j$ are never met. In this case, there is guaranteed to be a parameter $\hat{W}^2_{i,j} \neq 0$, setting the value of $f_i(\hat{\theta}, z^n)$ when the conditions of $R_j$ are met. If $R_j$ is never active for any $z^n$, then $\hat{W}^2_{i,j}$ does not affect the reconstruction loss $L_R$.

- **There is a condition of $R_j$ that we can remove without affecting the output of $f_{\mathcal{R}}$ on any example.** This condition is represented as a nonzero element in $\hat{W}^1_{j,k}$ (and also in $b_j$, which is not necessary for the proof). By construction this parameter does not affect the reconstruction loss.

In both cases, the gradient with respect to the nonzero parameter ($\hat{W}^2_{i,j}$ and $\hat{W}^1_{j,k}$) is set only by the regularizer. The right and left derivatives have the same sign, violating the conditions of strict coordinate-wise local optimality. $\qquad\square$

## B.8   Proof of Theorem 2

To prove Theorem 2, we will analyze the left and right derivatives of the loss function (eq. 17) with respect to each parameter in the compiled parameters $\hat{\theta}$, and show that the left derivative is positive and the right derivative is negative, satisfying the conditions of Proposition 1. The left and right derivatives of the loss function with respect to $w$, evaluated at $\hat{\theta}$, can be written as follows.

$$\frac{\partial^-}{\partial w} L(\hat{\theta}, \mathcal{D}, \alpha) = \frac{\partial^-}{\partial w}\left(\alpha|w|\right) + \frac{\partial^-}{\partial w}\left(\sum_n L_R(\hat{\theta}, z^n)\right) \tag{18}$$

$$\frac{\partial^+}{\partial w} L(\hat{\theta}, \mathcal{D}, \alpha) = \frac{\partial^+}{\partial w}\left(\alpha|w|\right) + \frac{\partial^+}{\partial w}\left(\sum_n L_R(\hat{\theta}, z^n)\right) \tag{19}$$

We can analyze the terms of these derivatives related to the regularizer and the reconstruction loss separately. First, we consider the regularizer term.

$$\frac{\partial^-}{\partial w}\left(\alpha|w|\right) = \begin{cases} -\alpha, & w \leq 0 \\ \alpha, & w > 0 \end{cases} \tag{20}$$

$$\frac{\partial^+}{\partial w}\left(\alpha|w|\right) = \begin{cases} -\alpha, & w < 0 \\ \alpha, & w \geq 0 \end{cases} \tag{21}$$

Next, let us consider the term related to the reconstruction loss. Let $\hat{\theta}^\epsilon_w$ be a set of parameters equal to $\hat{\theta}$, but with $\epsilon$ added to parameter $w$, e.g. $\hat{\theta}^\epsilon_{b^1_j} = \langle \hat{W}^1, \hat{b}^1 + \epsilon \cdot e_j, \hat{W}^2\rangle$, where $e_j$ is a standard basis vector with a 1 at position $j$ and $|\epsilon| < 1$. The left derivative can be written as follows,

$$\frac{\partial^-}{\partial w}\left(\sum_n L_R(\hat{\theta}, z^n)\right) = \lim_{\epsilon \to 0^-} \frac{1}{\epsilon}\left(\sum_n L_R(\hat{\theta}^\epsilon_w, z^n) - \sum_n L_R(\hat{\theta}, z^n)\right) \tag{22}$$

$$= \lim_{\epsilon \to 0^-} \frac{1}{\epsilon}\sum_n L_R(\hat{\theta}^\epsilon_w, z^n), \tag{23}$$

using $L_R(\hat{\theta}, z^n) = 0$ per eq. 16. The right derivative can be obtained similarly.

There are several ways we can show that the conditions of Proposition 1 are met for a given parameter $w$, i.e. that the left derivative $\frac{\partial^-}{\partial w} L(\hat{\theta}, \mathcal{D}, \alpha)$ is negative and the right derivative $\frac{\partial^+}{\partial w} L(\hat{\theta}, \mathcal{D}, \alpha)$ is positive.

**Lemma 1.** *If there exists some $d > 0$ such that $\sum_n L_R(\hat{\theta}^\epsilon_w, z^n) \geq |\epsilon|$ for all $\epsilon \in (-d, d)$, then the conditions of Proposition 1 are satisfied for $w$.*

*Proof.* The left and right derivatives of the reconstruction loss term are:

$$\lim_{\epsilon \to 0^-} \frac{1}{\epsilon} \sum_n L_R(\hat{\theta}_w^\epsilon, z^n) \leq \lim_{\epsilon \to 0^-} \frac{1}{\epsilon} |\epsilon| = -1 \tag{24}$$

$$\lim_{\epsilon \to 0^+} \frac{1}{\epsilon} \sum_n L_R(\hat{\theta}_w^\epsilon, z^n) \geq \lim_{\epsilon \to 0^+} \frac{1}{\epsilon} |\epsilon| = 1 \tag{25}$$

$$\tag{26}$$

Because $\alpha < 1$, for any value of $w$ we have:

$$\frac{\partial^-}{\partial w} L(\hat{\theta}, \mathcal{D}, \alpha) \leq -1 + \alpha < 0 \tag{27}$$

$$\frac{\partial^+}{\partial w} L(\hat{\theta}, \mathcal{D}, \alpha) \geq 1 - \alpha > 0. \tag{28}$$

Therefore the conditions of Proposition 1 are satisfied. □

**Lemma 2.** *If $w = 0$, then the conditions of Proposition 1 are satisfied for $w$.*

*Proof.* Because the reconstruction loss is zero at $\hat{\theta}$ and non-negative everywhere, its left and right derivatives are guaranteed to be $\leq 0$ and $\geq 0$, respectively. Therefore:

$$\frac{\partial^-}{\partial w} L(\hat{\theta}, \mathcal{D}, \alpha) \leq 0 - \alpha < 0 \tag{29}$$

$$\frac{\partial^+}{\partial w} L(\hat{\theta}, \mathcal{D}, \alpha) \geq 0 + \alpha > 0. \tag{30}$$

Therefore the conditions of Proposition 1 are satisfied. □

**Lemma 3.** *If $w > 0$ and there exists some $d > 0$ such that $\sum_n L_R(\hat{\theta}_w^\epsilon, z^n) \geq |\epsilon|$ for all $\epsilon \in (-d, 0)$, then the conditions of Proposition 1 are satisfied for $w$.*

*Proof.* This case therefore combines the reasoning of Lemmas 1 (for the left derivative) and 2 (for the right derivative).

$$\frac{\partial^-}{\partial w} L(\hat{\theta}, \mathcal{D}, \alpha) \leq -1 + \alpha < 0 \tag{31}$$

$$\frac{\partial^+}{\partial w} L(\hat{\theta}, \mathcal{D}, \alpha) \geq 0 + \alpha > 0. \tag{32}$$

Therefore the conditions of Proposition 1 are satisfied. □

*Remark.* Compared to Lemma 1, this Lemma combines a stronger condition on the sign of $w$ with a looser condition on the reconstruction loss, which in this case must grow with the perturbation $\epsilon$ only for $\epsilon < 0$.

We prove lemmas separately for each group of parameters, $\hat{W}^2$, $\hat{b}^1$, and $\hat{W}^1$, showing that the conditions of Lemma 1, 2, or 3 are met for every parameter in $\hat{\theta}$.

**Lemma 4.** *Each $\hat{W}_{i,j}^2$ satisfies the conditions of Propostion 1.*

*Proof.*

$$L_R(\hat{\theta}^\epsilon_{W^2_{i,j}}, z^n) = \sum_{i'} |f_{i'}(\hat{\theta}^\epsilon_{W^2_{i,j}}, V) - f_{i'}(\hat{\theta}, V)| \tag{33}$$

$$= \sum_{i'} \left| \left[ z^n_{i'} + \sum_{j'} (\hat{W}^2_{i,j'} + \epsilon \times [\![j = j' \wedge i = i']\!]) h_{j'}(\hat{\theta}, z^n) \right] - \tag{34}$$

$$\left[ z^n_{i'} + \sum_{j'} \hat{W}^2_{i,j'} h_{j'}(\hat{\theta}, z^n) \right] \right|$$

$$= |\epsilon \times h_j(\hat{\theta}, z^n)|. \tag{35}$$

Lines 33 and 34 plug in the definitions of $L_R$ (eq. 15) and $f_i$ (eq. 7) respectively, and line 35 follows from algebra.

Recall that $h_j(\hat{\theta}, z^n) \in \{0, 1\}$ for all $(j, n)$, per eq. 12. For all $n$ where $h_j(\hat{\theta}, z^n) = 0$, the value of $L_R(\hat{\theta}^\epsilon_{W^2_{i,j}}, z^n) = 0$ for all $i$, but by minimality there must be some $z^n$ for which $h_j(\hat{\theta}, z^n) = 1$ or else rule $j$ could be removed. Therefore:

$$\sum_n L_R(\hat{\theta}^\epsilon_{W^2_{i,j}}, z^n) = \sum_n |\epsilon \times h_j(\hat{\theta}, z^n)| \geq |\epsilon|. \tag{36}$$

Therefore the conditions of Lemma 1 are met. $\qquad\square$

**Lemma 5.** *Each $\hat{b}^1_j$ satisfies the conditions of Propostion 1.*

*Proof.* We can derive the following by substituting the definitions of $L_R$ (eq. 15) and $f_i$ (eq. 7), $h_j$ (eq. 6), and $g_j$ (eq. 5), and simplifying the resulting expression, for $\epsilon < 1$:

$$L_R(\hat{\theta}^\epsilon_{b^1_j}, z^n) = \sum_i |f_i(\hat{\theta}^\epsilon_{b^1_j}, z^n) - f_i(\hat{\theta}, z^n)| \tag{37}$$

$$= \sum_i |\hat{W}^2_{i,j} \times \epsilon \times [\![g_j(\hat{\theta}, z^n) + \epsilon \in (0, 1)]\!]| \tag{38}$$

$$= |\epsilon| \times [\![g_j(\hat{\theta}, z^n) + \epsilon \in (0, 1)]\!] \times \sum_i |W^2_{i,j}|. \tag{39}$$

$$\tag{40}$$

By construction of $W^2$, for all $j$ there exists $|W^2_{i,j}| = 1$, i.e. there is a non-zero entry in every column of $W^2$ (see eq. 10). Therefore, $\sum_i |W^2_{i,j}| \geq 1$, and:

$$L_R(\hat{\theta}^\epsilon_{b^1_j}, z^n) \geq |\epsilon| \times [\![g_j(\hat{\theta}, z^n) + \epsilon \in (0, 1)]\!]. \tag{41}$$

By construction of $\hat{\theta}$, for all $n$, $g_j(\hat{\theta}, z^n) \in \{1, 0, -1, \cdots\}$ (per eq. 11). Therefore we can enumerate the possible cases:

$$L_R(\hat{\theta}^\epsilon_{b^1_j}, z^n) \geq \begin{cases} |\epsilon|, & g_j(\hat{\theta}, z^n) = 0 \wedge \epsilon > 0 \\ |\epsilon|, & g_j(\hat{\theta}, z^n) = 1 \wedge \epsilon < 0 \\ 0, & \text{otherwise.} \end{cases} \tag{42}$$

By the minimality criteria, for all $j$ there exists some $n$ where $g_j(\hat{\theta}, z^n) = 0$, where $N - 1$ conditions of rule $j$ are satisfied (see eq. 11). If not, some condition could be removed, violating minimality. Also, for all $j$ there exists some $n$ where $g_j(\hat{\theta}, z^n) = 1$, where all $N$ conditions of rule $j$ are satisfied. If not, the rule could be removed, violating minimality. Therefore:

$$\sum_n L_R(\hat{\theta}^\epsilon_{b^1_j}, z^n) \geq |\epsilon|. \tag{43}$$

Therefore, the conditions of Lemma 1 are met. $\qquad\square$

**Lemma 6.** *Each $\hat{W}_{j,k}^1$ satisfies the conditions of Propostion 1.*

*Proof.* The analysis is similar to that of $\hat{b}_j^1$, and we can reuse the result of eq. 41.

$$L_R(\hat{\theta}_{W_{j,k}^1}^\epsilon, z^n) = \sum_i |f_i(\hat{\theta}_{W_{j,k}^1}^\epsilon, z^n) - f_i(\hat{\theta}, z^n)| \tag{44}$$

$$= \sum_i |z_k^n \times \hat{W}_{i,j}^2 \times \epsilon \times [\![g_j(\hat{\theta}, z^n) + \epsilon \in (0,1)]\!]| \tag{45}$$

$$= |z_k^n| \times \underbrace{\sum_i |\hat{W}_{i,j}^2 \times \epsilon \times [\![g_j(\hat{\theta}, z^n) + \epsilon \in (0,1)]\!]|}_{L_R(\hat{\theta}_{b_j^1}^\epsilon, z^n)} \tag{46}$$

$$\geq |z_k^n| \times |\epsilon| \times [\![g_j(\hat{\theta}, z^n) + \epsilon \in (0,1)]\!]. \tag{47}$$

Recall that $W_{j,k}^1 \in \{0,1\}$ (eq. 8) and $z_k^n \in \{0,1\}$ (eq. 1). There are two cases to consider.

**Case 1:** $W_{j,k}^1 = 1$. By minimality, for all $k$ there exists some $n$ such that $z_k^n = 1$ and $g_j(\hat{\theta}, z^n) = 1$, i.e. where rule $R_j$ is satisfied by $z^n$, and thus necessarily the constraint represented by $W_{j,k}^1$ is satisfied by the input. However, minimality does not guarantee that there exists some $n$ such that $z_k^n = 1$ and $g_j(\hat{\theta}, z^n) = 0$. Note that if $\epsilon < 0$, then $|z_k^n| \times |\epsilon| \times [\![g_j(\hat{\theta}, z^n) + \epsilon \in (0,1)]\!] = |\epsilon|$ if $z_k^n = 1$ and $g_j(\hat{\theta}, z^n) = 1$, which is the case for some $n$. Therefore:

$$\sum_n L_R(\hat{\theta}_{W_{j,k}^1}^\epsilon, z^n) = \sum_n |z_k^n| \times |\epsilon| \times [\![g_j(\hat{\theta}, z^n) + \epsilon \in (0,1)]\!] \geq \begin{cases} |\epsilon|, & \epsilon < 0 \\ 0, & \epsilon \geq 0 \end{cases} \tag{48}$$

Since $W_{j,k}^1 > 0$, the conditions of Lemma 3 are met.

**Case 2:** $W_{j,k}^1 = 0$. The conditions of Lemma 2 are met. $\qquad\square$

## C   Learnability Analysis Details

In this section, we provide additional discussion and details related to trace supervision (§C.1) and the procedure for determining the minimal version of a program with respect to a training set (§C.2).

### C.1   Procedure for Training with Trace Supervision

Given a program $P$ and a set of model inputs $\mathcal{X}$, we can run $P$ for every input in $\mathcal{X}$, and extract the variable assignments at the input and output of every sub-layer, for every position. In our experiments, for simplicity, we focus on training the MLP parameters. After using the ALTA interpreter to collect pairs of variable assignments at the input and output of the MLP layers, we map these assignments to vectors, using the mapping described in §2. Finally, we can train a MLP layer based on these pairs of input and output vectors. Specifically, we use an L2 loss to encourage the MLP to produce the output vector given the input. We then use the ALTA compiler to compile parameters other than those used for the MLP layer. By combining the learned MLP parameters with those provided by the compiler, we have a full set of Transformer parameters. See §D.1 for the hyperparameters and training details for the parity task.

For future work, it may be possible to provide supervision from execution traces in a way that makes fewer assumptions about how variables are encoded in the residual stream, i.e. by encouraging the residual stream to encode such values in a way that allows them to be accurately predicted from the residual stream with a linear classifier. This would enable training models with intermediate supervision without any compiled parameters, but is out of scope for this work. It would then be interesting to assess whether there is any potential for transfer learning from training with such supervision in cases where a ground truth program is known to cases where a ground truth program is unknown or not feasible to express.

### C.2 Procedure for Determining Minimal Versions of Programs

Given a program $P$ and set of model inputs $\mathcal{X}$, we can determine the *minimal version* of $P$ with respect to $\mathcal{X}$. For simplicity, we focus on the set of transition rules and the input embedding operations for this analysis. First, as motivated by our theoretical analysis in Appendix B, we remove any transition rules which are never satisfied when running $P$ on the inputs in $\mathcal{X}$. While the formal definition of a *minimal rule set* also puts conditions on the constraints of satisfied rules, we assume that there are no unnecessary constraints to simplify our analysis. Second, we analyze the set of input IDs and positions seen when executing $P$ over $\mathcal{X}$. We restrict the variable initialization functions of the minimal program to output default values for variables outside of this minimal set of observed token IDs and positions.

## D Program Details and Additional Results

In this section we provide program details and additional results. Some program listings omit the definitions of various constants and helper functions for brevity, and we refer the reader to the open-source code for the complete program specifications for all programs discussed in this paper.

### D.1 Parity

**Programs** Here we provide the ALTA code for the parity programs that we study. The code for the *Sequential (Relative)* program is in Figure 8 and the code for the *Sum + Modulo* program is in Figure 7. The code for the *Sequential (Absolute)* program was given in Figure 2.

```
vars = {
    "parity": var(range=2),
    "start": numeric_var(input_init_fn=lambda x: float(x == START),
                         values=(0, 1)),
    "start_or_one": var(range=2, input_init_fn=lambda x: x in {1, START}),
    "query": var(range=2, default=1),
}

attention_heads = {
    "x": qkv("query", "start_or_one", "start",
             output_spec=numeric_var(values=BUCKETS)),
}

def ffn_fn(z):
  num_ones = round(1 / z["x"]) - 1
  z["parity"] = int(num_ones % 2 != 0)

return program_spec(
    variables=vars, heads=attention_heads, ffn_fn=ffn_fn,
    output_name="parity", input_range=3, position_range=None
)
```

Figure 7: Program for computing parity using a sum and modulo operation.

**Model Interface** Each program expects the input to be a binary sequence. The *Sum + Modulo* and *Sequential (Relative)* programs additionally expect a `START` token to be prepended to the sequence. For all programs, the final token in the output sequence contains the parity of the input sequences.

**Model Sizes** The compiled model for the *Sequential (Relative)* program has an MLP width (i.e., number of transition rules) of 7, and 2 attention heads. The number of transition rules for the minimal version of the *Sequential (Absolute)* program depends on the maximum input length in the training set, and the number of transition rules for the minimal version of the *Sum + Modulo* program depends on the maximum number of ones in the training set.

```
variables = {
    "parity": var(range=2, input_init_fn=lambda x: 0 if x == START else x),
    "done": var(range=2, input_init_fn=lambda x: x == START),
}
attention_heads = {
    "parity_left": v_relative("parity", -1),
    "done_left": v_relative("done", -1),
}

def ffn_fn(z):
  if z["done"] != 1 and z["done_left"] == 1:
    z["parity"] = z["parity_left"] ^ z["parity"]
    z["done"] = 1

return program_spec(
    variables=variables, heads=attention_heads,ffn_fn=ffn_fn,
    output_name="parity", input_range=3, position_range=None,
    halt=halt_spec("done", halt_value=1),
)
```

Figure 8: Program for computing parity sequentially using relative positions.

**Alternative Programs**   There is another algorithm for parity proposed by Chiang & Cholak (2022), which was inspired by algorithms for MLPs from Rumelhart et al. (1986). Similarly to the Sum + Modulo program, this algorithm first computes a sum operation using an attention head, but they avoid explicitly computing the modulo operation by instead using attention heads that separately attend to even and odd elements. This enables the algorithm to be implemented with a fixed set of parameters that is invariant to the maximum input length, aside from the positional encodings. However, we did not explore this algorithm because it is not possible to implement in a way where the minimal ALTA program is invariant to the maximum input length considered. This is because ALTA requires discretization of numerical variables in order to perform numerical computations, a current limitation of the framework. Implementing the computations required would therefore require a number of transition rules that scales with the maximum input length considered, similarly to how the Sum + Modulo program requires a number of buckets that scales with the maximum number of ones considered. Regardless, Chiang & Cholak (2022) showed this algorithm is difficult to learn in practice, and it also requires specialized positional encodings. Furthermore, their algorithm requires encoding parameters and activations with a degree of numerical precision that scales with the maximum input length. In contrast, the *Sequential (Relative)* program compiles to a Universal Transformer that only needs to encode 4 binary variables in the residual stream, and consists of only 7 transition rules, i.e., requires only 7 hidden MLP dimensions. However, the construction of Chiang & Cholak (2022) requires only 2 layers, where as the *Sequential (Relative)* program requires $N - 1$ layers, where $N$ is the maximum input length.

**Train and Test Sets**   The train and test sets are the same for all experiments (including both trace and end-to-end supervision). The train set consists of examples between lengths 0 and 20, and the test set contains examples between lengths 0 and 40. The sets include roughly an equal number of examples per number of ones.

**Trace Supervision Details**   Hyperparameters for training with trace supervision are listed in Table 1. All are standard hyperparameters for training neural networks except "Noise Std Dev." We added a small amount of Gaussian noise to the neural network input at training time to make it robust to numeric imprecision at inference time.

The *Sequential (Absolute)* and *Sum + Modulo* experiments used four hidden layers instead of the standard two, as we explored various hyperparameters to ensure the MLP had the capacity to fit the training set. Similarly, we used Adam (Diederik, 2014) for both experiments instead of Adafactor (Shazeer & Stern, 2018), as it helped fit the training set.

Table 1: Trace supervision hyperparameters.

| | Program | | |
|---|---|---|---|
| Hyperparameter | Sequential (Relative) | Sequential (Absolute) | Sum + Modulo |
| Hidden Layers | 2 | 4 | 4 |
| Hidden Layer Size | 128 | 4,096 | 4,096 |
| Batch Size | 256 | 256 | 256 |
| Steps | 50,000 | 50,000 | 400,000 |
| Learning Rate | 1e-2 | 1e-4 | 1e-4 |
| Activation Fn | ReLU | ReLU | ReLU |
| Optimization Fn | Adafactor | Adam | Adam |
| Noise Std Dev | 0.1 | 0.1 | 0.1 |

The *Sum + Modulo* program used for trace supervision differs slightly from the program in Figure 7. `num_ones` is stored as an intermediate categorical variable, as we found it easier for the MLP to learn to map a categorical variable to the correct parity output than a numeric variable.

**End-to-end Training Details** We trained transformers with various configurations using standard supervision — varying the number of layers, whether weight sharing is used, and the type of positional encoding. Constant in all standard supervision experiments are the following hyperparameters: embeddings with dimension 512, hidden layer sizes of 2048, 6 attention heads with dimension 64, an Adafactor optimization function, GeLU (Hendrycks & Gimpel, 2016) activation functions, a learning rate of 5e-4, 50,000 steps, and a byte vocabulary.

Table 2: Length generalization accuracy for Transformers trained with intermediate supervision on parity.

| | Accuracy | | |
|---|---|---|---|
| Program | Length $\leq 20$ | Length $> 20$ | Ones $> 20$ |
| Sequential (Relative) | 100% | 100% | 100% |
| Sequential (Absolute) | 100% | 52% | 51% |
| Sum + Modulo | 100% | 78% | 50% |

Table 3: Length generalization accuracy for Transformers trained with standard supervision on parity.

| | | | Accuracy | | |
|---|---|---|---|---|---|
| Layers | Weight Sharing | Positional Encoding | Length $\leq 20$ | Length $> 20$ | Ones $> 20$ |
| 8 | No | Relative | 100% | 60% | 51% |
| 8 | Yes | Relative | 100% | 65% | 52% |
| 40 | No | Relative | 100% | 55% | 51% |
| 40 | Yes | Relative | 100% | 54% | 48% |
| 8 | No | Absolute | 100% | 49% | 50% |
| 8 | Yes | Absolute | 100% | 50% | 52% |
| 40 | No | Absolute | 100% | 50% | 51% |
| 40 | Yes | Absolute | 100% | 52% | 51% |

**Results** Table 2 presents the accuracy for each intermediate supervision experiment and Table 3 presents the accuracy for each standard supervision configuration on different slices of the data. Figure 5 compares the intermediate and standard supervision results, showing that standard supervision exhibits behavior similar to the *Sum + Modulo* program.

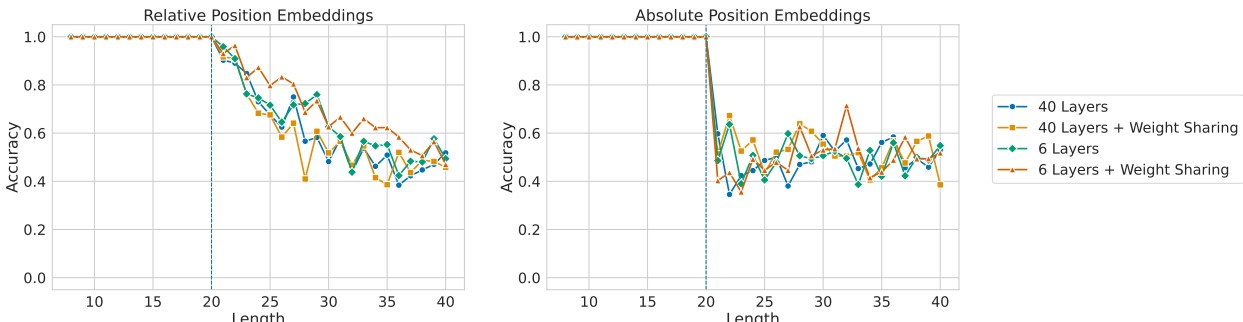

Figure 9: Accuracy by length for Transformers trained with standard supervision. There is some length generalization with relative position embeddings (left), but none with absolute (right).

Figures 9, 10, and 11 break down the standard supervision results in more detail.

Figure 9 shows that there is no length generalization when using absolute positional embeddings, while with relative positional embeddings there is some length generalization which decreases as length increases.

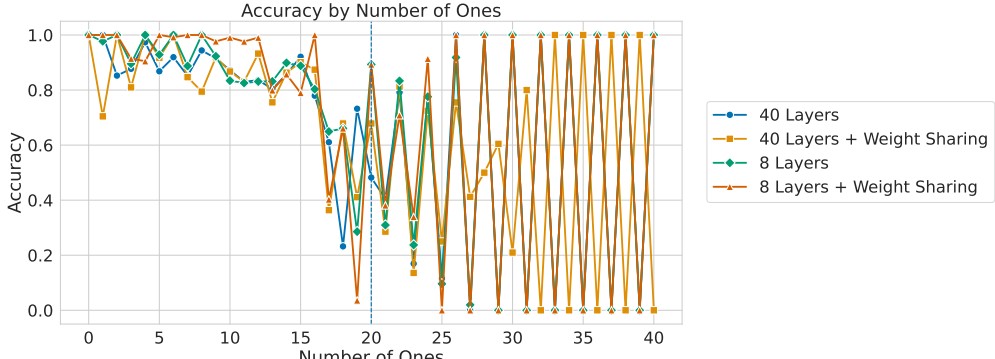

Figure 10: Accuracy by number of ones for Transformers trained with standard supervision using relative position embeddings. There is no generalization beyond 20 ones, which is the maximum number seen during training.

Figure 10 shows that the length generalization we observe with relative positional embeddings is due entirely to longer examples sometimes containing the same number of ones as shorter examples in the training distribution, consistent with results in Anil et al. (2022). After 20 ones, accuracy starts oscillating between 0 and 1. The oscillations are due to the model predicting either 0 or 1 for all examples once examples contain more ones than were seen during training, which is shown in Figure 11.

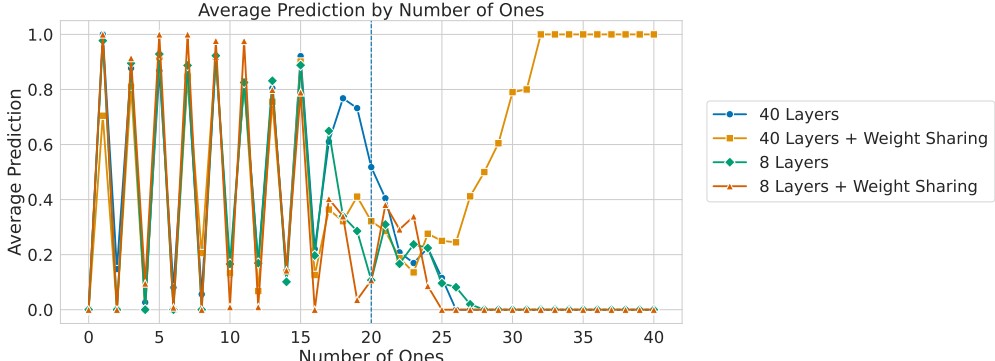

Figure 11: Average prediction by number of ones for Transformers trained with standard supervision using relative position embeddings. Once examples have slightly more than 20 ones, the maximum number seen during training, the models resort to predicting either 0 or 1 for all examples.

## D.2 Addition

While the addition task has been previously studied in the context of decoder-only Transformers, it is also straightforward to compute addition in an encoder-only Transformer. We give an example input and output sequence pair in Table 4.

Table 4: Example input and output sequences for addition task.

| Input Sequence | START | 3 | 4 | + | 5 | 6 | END |
|---|---|---|---|---|---|---|---|
| Output Sequence | PAD | PAD | PAD | PAD | 9 | 0 | PAD |

Our program for computing addition is given in Figure 13. The algorithm initializes a pointer to the first digit of each input, and a pointer to an output buffer. The algorithm then iterates over the digits, adding the current digits, adding the value to the output buffer, and updating the carry value. When the last digit has been processed, the final carry value is added to the output buffer.

The minimal compiled MLP width (i.e., number of transition rules) is 884, and the program uses 6 attention heads. The program requires $N + 2$ layers, where $N$ is the number of digits in the larger of the two inputs.

```python
def init(x):
  # Initializes pointers.
  if x["token_right"] == END_TOKEN:
    x["ptr_b"] = 1
    x["ptr_out"] = 1
  if x["token_right"] == ADD_TOKEN:
    x["ptr_a"] = 1

def iterate(x):
  # Execute one step of addition.
  raw_sum = x["value_carry"] + x["ptr_a_token"] + x["ptr_b_token"]
  if x["ptr_out"]:
    x["value_out"] = raw_sum % 10
  x["value_carry"] = raw_sum // 10
  # Move all pointers to the left.
  # Attention heads attending to the right will be undefined.
  if x["token"] != END_TOKEN:
    x["ptr_out"] = x["ptr_out_right"]
    x["ptr_a"] = x["ptr_a_right"]
    x["ptr_b"] = x["ptr_b_right"]

def finalize(x):
  # Finalize output by adding the final carry to the output.
  if x["ptr_out"]:
    x["value_out"] = x["value_carry"]
  x["step"] = STEP_DONE

def ffn_fn(x):
  if x["step"] == STEP_INIT:
    init(x)
    x["step"] = STEP_ITERATE
  elif x["step"] == STEP_ITERATE:
    if x["ptr_a_token"] == START_TOKEN:
      x["step"] = STEP_FINALIZE
    else:
      iterate(x)
  elif x["step"] == STEP_FINALIZE:
    finalize(x)
```

Figure 12: MLP function for addition program.

```python
variables = {
  "token": pb.var(INPUT_RANGE, input_init_fn=lambda x: x),
  # This variable tracks the current processing step.
  "step": pb.var(NUM_STEPS),
  # These are pointers to which digit is currently being processed.
  # They are '1' at the position of the current digit to process, and '0'
  # otherwise.
  "ptr_a": pb.var(2),
  "ptr_b": pb.var(2),
  # This pointer is '1' at the position to write the next output to,
  # and '0' otherwise.
  "ptr_out": pb.var(2),
  # This tracks the "carry" value form the previous iteration.
  "value_carry": pb.var(10),
  # This tracks the final output value for a given digit.
  "value_out": pb.var(10),
  # Static variables used as attention query.
  "one": pb.var(var_range=2, default=1),
}
attention_heads = {
  # For these relative attention heads, we always want to attend to the
  # position immediately to the right.
  "token_right": v_relative("token", 1),
  "ptr_a_right": v_relative("ptr_a", 1),
  "ptr_b_right": v_relative("ptr_b", 1),
  "ptr_out_right": v_relative("ptr_out", 1),
  # For these attention heads, we want to attend to the positions associated
  # with the current pointers.
  "ptr_a_token": qkv("one", "ptr_a", "token"),
  "ptr_b_token": qkv("one", "ptr_b", "token"),
}
return program_spec(
  variables=variables, heads=attention_heads, ffn_fn=ffn_fn,
  output_name="value_out", input_range=INPUT_RANGE, position_range=None,
  halt_spec=pb.halt_spec("step", halt_value=STEP_DONE),
)
```

Figure 13: Program for adding two positive integers. The MLP function is defined in Figure 12.

### D.3  SUBLEQ

Our ALTA program for implementing a SUBLEQ interpreter is given in Figure 15. The input tokens define the set of memory registers, and program execution starts at position 0. SUBLEQ stands for SUBtract and branch if Less-than or EQual to zero. Commands in SUBLEQ are specified by three memory addresses $A$, $B$, and $C$. Executing a command consists of subtracting the value at memory address $A$ from the value at address $B$, and writing the result to address $B$. If the result is less than or equal zero, then the program jumps to the command at address $C$, or will halt if $C < 0$. Giannou et al. (2023) showed how a Looped Transformer can implement an interpreter for a restricted form of SUBLEQ that did not allow self-modifying code, i.e., memory registers specifying SUBLEQ instructions could not be modified during program execution. Our program implements a SUBLEQ interpreter without such restrictions, i.e. without differentiating between program and memory registers. Additionally, our implementation executes 1 SUBLEQ instruction every 3 layers, as opposed to every 9 layers for the implementation of Giannou et al. (2023). However, our approach requires $\mathcal{O}(N^3)$ MLP hidden dimensions (i.e., transition rules), where $N$ is the number of possible memory values. The construction of Giannou et al. (2023) requires only $\mathcal{O}(logN)$ MLP hidden dimensions.

Our compiled models use 5 attention heads. Each token in the input and output of our model corresponds to a single register, with the input and output sequences encoding the initial and final values of the registers, respectively. We verified the correctness of our program and compiled models using unit tests involving simple SUBLEQ programs.

```python
def encode(value):
  # Encodes a register value as positive integer.
  return value - MIN_VALUE

def decode(value):
  # Decodes a register value from positive integer.
  return value + MIN_VALUE

def _update_position(z, position_a):
  z["position_a"] = position_a
  z["position_b"] = position_a + 1
  z["position_c"] = position_a + 2

def ffn_fn(z):
  if z["state"] == STATE_1:
    if decode(z["a"]) < 0 or decode(z["b"]) < 0:
      # 'a' or 'b' are not valid register positions.
      z["state"] = STATE_DONE
      return
    z["state"] = STATE_2
  elif z["state"] == STATE_2:
    # Compute mem[b] - mem[a].
    mem_b = decode(z["mem_b"]) - decode(z["mem_a"])
    z["jump"] = int(mem_b <= 0)

    # Update memory value at position 'b'.
    if z["position"] == z["b"]:
      z["mem"] = encode(mem_b)

    z["state"] = STATE_3
  elif z["state"] == STATE_3:
    # Determine next instruction.
    if z["jump"]:
      # Jump to instruction 'c'.
      if decode(z["c"]) < 0:
        # Break if 'c' is negative.
        z["state"] = STATE_DONE
      else:
        _update_position(z, z["c"])
        z["state"] = STATE_1
    else:
      # Proceed to next instruction.
      _update_position(z, z["position_a"] + 3)
      z["state"] = STATE_1
```

Figure 14: MLP function for interpreting SUBLEQ instructions.

```
mem_range = (MAX_VALUE - MIN_VALUE) + 1
variables = {
    # Value of register.
    "mem": var(mem_range, input_init_fn=lambda x: x),
    # Position of register.
    "pos": var(mem_range, position_init_fn=encode),
    # Position of current instruction.
    "pos_a": var(mem_range, default=encode(0)),
    "pos_b": var(mem_range, default=encode(1)),
    "pos_c": var(mem_range, default=encode(2)),
    # Program state.
    "state": var(NUM_STATES),
    # Whether to jump at next instruction.
    "jump": var(2),
}

attention_heads = {
    # Values of registers at 'pos_a', 'pos_b', and 'pos_c'.
    "a": qkv("pos_a", "pos", "mem"),
    "b": qkv("pos_b", "pos", "mem"),
    "c": qkv("pos_c", "pos", "mem"),
    # Value of registers at 'a' and 'b'.
    "mem_a": qkv("a", "pos", "mem"),
    "mem_b": qkv("b", "pos", "mem"),
}

return program_spec(
    variables=variables, heads=attention_heads, ffn_fn=ffn_fn,
    output_name="mem", input_range=mem_range, pos_range=NUM_POSITIONS,
    halt=halt_spec("state", halt_value=STATE_DONE),
)
```

Figure 15: Program for interpreting SUBLEQ instructions. The MLP function is specified in Figure 14.

Table 5: Example of shift-reduce parsing for SCAN, for the input "jump twice". Our ALTA program represents the state of the stack and parse tree using a variable number of input tokens.

| Action | Stack | Parse Tree | Input Buffer |
|---|---|---|---|
| Initialize | $\langle\,\rangle$ | $\langle\,\rangle$ | $\langle$ `jump`, `twice` $\rangle$ |
| Shift | $\langle$ `jump` $\rangle$ | $\langle\,\rangle$ | $\langle$ `twice` $\rangle$ |
| Reduce | $\langle$ `NT` $\rangle$ | $\langle$ `NT` $\rightarrow$ `jump` $\rangle$ | $\langle$ `twice` $\rangle$ |
| Shift | $\langle$ `NT`, `twice` $\rangle$ | $\langle$ `NT` $\rightarrow$ `jump` $\rangle$ | $\langle\,\rangle$ |
| Reduce | $\langle$ `NT` $\rangle$ | $\langle$ `NT` $\rightarrow$ `jump`, `NT` $\rightarrow$ `NT twice` $\rangle$ | $\langle\,\rangle$ |

## D.4 SCAN

**Background**  The SCAN suite of compositional generalization tasks (Lake & Baroni, 2018) require mapping natural language commands (e.g., "*jump twice and look left*") to action sequences (e.g., `JUMP JUMP LTURN LOOK`). The suite is inspired by the linguistic notion of systematic compositionality, i.e., the ability to recombine a finite set of elements in novel ways (Chomsky, 1957; Montague, 1970). Certain splits have been shown to be challenging for Transformer-based models (Keysers et al., 2020; Furrer et al., 2020; Qiu et al., 2022b; Kazemnejad et al., 2024), especially the length split and the Maximum Compound Divergence (MCD) splits proposed by Furrer et al. (2020). Notably, no Transformer-based model has been shown to reliably solve these tasks, without relying on some symbolic decomposition of the task (Zhou et al., 2023a) or training data augmented by a symbolic system (Qiu et al., 2022a). Other successful solutions have also involved symbolic parsing of some form (Shaw et al., 2021; Chen et al., 2020; Herzig & Berant, 2021). While prior work has studied the expressivity of various classes of Transformers with respect to formal languages, it has primarily focused on recognizing and generating Dyck languages (Yao et al., 2021; Bhattamishra et al., 2020; Hahn, 2020; Weiss et al., 2021; Ebrahimi et al., 2020), and thus has not previously shown a constructive demonstration of how a Transformer can solve the SCAN task.

**Program**  Our approach follows Shaw et al. (2021) in formalizing the SCAN task as translation given a quasi-synchronous context-free grammar. Notably, the SCAN grammar is unambiguous and can be parsed in linear time. First, the program executes a shift-reduce parse of the input sequence, representing the parse as a tree. Second, the ALTA program decodes the output sequence by traversing the parse tree. The program represents the necessary variable-length data structures (a stack, parse tree, and buffer) using a variable number of input tokens. We include additional "memory" tokens in the input to ensure there are a sufficient number of tokens to represent these structures. We give an example of shift-reduce parsing for SCAN in Table 5.

Similarly to our approach to the addition task, while SCAN has commonly been studied using encoder-decoder or decoder-only Transformers, we can also represent the task using an encoder-only Transformer. We ensure there is a sufficient number of "memory" tokens so that the entire output sequences can be represented in the encoder output, even if the output sequence exceeds the length of the input sequence.

In Figure 17 we show a program for parsing SCAN inputs using a shift-reduce parser. This program is a fragment of the overall SCAN program, which also decodes the output from the parsed representation, but is too long to include in this paper. We refer the reader to our open-source code for the full SCAN program.

**Program Size**  The minimal version of the SCAN program compiles to a Transformer encoder with fewer than 2,000 MLP dimensions and 13 attention heads. The number of layers required scales with the input length and number of tree traversal operations to decode the output from the parse tree. All examples in the dataset can be parsed with fewer than 512 layers. However, it is likely that more layer-efficient implementations exist.

**End-to-end Training Details**  We trained *encoder-decoder* Transformers end-to-end on the SCAN length and MCD splits. We varied the number of layers, whether weight sharing was used, and the type of positional

```python
def shift_stack_pointers(z, stack_pointer_offset):
  new_stack_pointer_0 = z["stack_pointer_0"] + stack_pointer_offset
  z["stack_pointer_0"] = new_stack_pointer_0
  z["stack_pointer_1"] = new_stack_pointer_0 - 1
  z["stack_pointer_2"] = new_stack_pointer_0 - 2
  z["stack_pointer_3"] = new_stack_pointer_0 - 3

def reduce(z, matched_rule):
  # Pop RHS elements and add LHS nonterminal to stack.
  if z["position"] == (z["stack_pointer_0"] - rule_len(matched_rule)):
    z["symbol_id"] = rule_lhs_id(matched_rule)
  shift_stack_pointers(z, 1 - rule_len(matched_rule))
  # Add rule to parse tree.
  if z["position"] == z["tree_pointer"]:
    # Use 1-indexing to reserve 0 for no rule.
    z["rule_id"] = rule_id(matched_rule)
  z["tree_pointer"] += 1

def shift(z):
  # Shift the next token to the stack.
  if z["position"] == z["stack_pointer_0"]:
    z["symbol_id"] = get_symbol_id(z["input_pointer_token_id"])
  shift_stack_pointers(z, 1)
  z["input_pointer"] += 1

def ffn_fn(z):
  if not z["done"]:
    # Check if top-3 stack symbols (and 1 lookahead token) match any rule.
    matched_rule = maybe_match_rule(
        z["input_pointer_token_id"],
        z["stack_symbol_1"],
        z["stack_symbol_2"],
        z["stack_symbol_3"],
    )
    if matched_rule is not None:
      reduce(z, matched_rule)
    else:
      # Check if parsing is complete.
      if z["input_pointer_token_id"] == EOS_ID:
        z["done"] = 1
      else:
        shift(z)
```

Figure 16: MLP function for parsing SCAN.

encodings used in the encoder. When using weight sharing, we trained with up to 256 encoder layers and 256 decoder layers (as our ALTA program requires at most 512 total layers). Without weight sharing, we only trained with up to 64 encoder layers and 64 decoder layers due to memory constraints.

Constant in all experiments were the following hyperparameters: embeddings with dimension 128, hidden layer sizes of 512, 8 attention heads with dimension 128, an Adafactor optimization function, GeLU activation functions, a learning rate of 5e-4, 100,000 steps, and a SentencePiece vocabulary (Kudo & Richardson, 2018).

**Results** Figure 18 plots test accuracy when using weight sharing and relative positional encodings. Accuracy does not improve as the number of layers increases. (If anything, there is a slight inverse relationship between number of layers and test accuracy.) This is the case in all experiments, regardless of the type of positional encoding used in the encoder and whether weight sharing is used. (See Table 6 for all results.)

The fact that increasing the number of layers does not improve generalization indicates that when trained with standard supervision, Transformers are unable to take advantage of extra layers to learn a function consistent with the sequential ALTA program that generalizes perfectly. Notably, all end-to-end experiments fit the training set, even with just two layers, so there exists some other algorithm that fits the training set that requires at most two layers. We speculate that in all cases, the end-to-end supervised Transformers are unable to learn the sequential algorithm with perfect generalization because of a bias towards learning

```
variables = {
    "token": var(NUM_INPUT_TOKENS, input_init_fn=lambda x: x),
    "position": var(NUM_POSITIONS, position_init_fn=lambda x: x),
    # Whether parsing is complete.
    "done": var(2),
    # Pointer to the next stack position, and then the top 3 elements on
    # the stack.
    "stack_pointer_0": var(NUM_POSITIONS, default=STACK_OFFSET),
    "stack_pointer_1": var(NUM_POSITIONS, default=STACK_OFFSET - 1),
    "stack_pointer_2": var(NUM_POSITIONS, default=STACK_OFFSET - 2),
    "stack_pointer_3": var(NUM_POSITIONS, default=STACK_OFFSET - 3),
    # Pointer to write the next rule to.
    "tree_pointer": var(NUM_POSITIONS, default=TREE_OFFSET),
    # Pointer to the next input token to process.
    "input_pointer": var(NUM_POSITIONS, default=INPUT_OFFSET),
    # Stores index of associated parsing rule.
    "rule_id": var(NUM_RULES),
    # Stores symbol ID associated with stack element.
    "symbol_id": var(NUM_SYMBOLS),
}

heads = {
    # Get token at input pointer.
    "input_pointer_token_id": qkv("input_pointer", "position", "token"),
    # Get top 3 symbols on stack.
    "stack_symbol_1": qkv("stack_pointer_1", "position", "symbol_id"),
    "stack_symbol_2": qkv("stack_pointer_2", "position", "symbol_id"),
    "stack_symbol_3": qkv("stack_pointer_3", "position", "symbol_id"),
}

return program_spec(
    variables=variables, heads=heads, ffn_fn=ffn_fn,
    output_name="rule_id",
    input_range=NUM_INPUT_TOKENS,
    position_range=NUM_POSITIONS,
)
```

Figure 17: Program for parsing SCAN. The MLP function is defined in Figure 16.

algorithms that require fewer layers to execute. These results are consistent with our results on Parity, in which Transformers trained with end-to-end supervision do not learn the sequential algorithm with perfect generalization, instead seeming to mimic an algorithm that requires just one layer (see §D.1).

Similarly, prior work has generally not shown a consistent benefit from increasing model size or number of layers on SCAN or other similar compositional generalization tasks when fine-tuning. Furrer et al. (2020) evaluated models of various sizes on SCAN and did not find that increasing size improves generalization. Ontanon et al. (2022) showed some improvement on the SCAN length split when increasing the number of layers with weight sharing, but only evaluated up to six layers. Qiu et al. (2022b) and Petty et al. (2024) systematically evaluated the impact of model size and depth, respectively, on compositional generalization (though not on SCAN) and found that when fine-tuning, scaling curves are flat or quickly saturate. However, these papers evaluated models with at most 118 layers, and only Ontanon et al. (2022) evaluated models with varying numbers of layers using weight sharing.

Table 6: Test accuracy for all Transformers trained with standard, end-to-end supervision on SCAN. Regardless of the configuration and split, increasing the number of layers does not improve test accuracy.

| | | | Number of Layers | | | | | | |
|---|---|---|---|---|---|---|---|---|---|
| Split | Weight Sharing | Encoder Positional Encoding | 2 | 6 | 16 | 32 | 64 | 128 | 256 |
| Length | Yes | Relative | 11% | 10% | 9% | 7% | 7% | 7% | 5% |
| Length | Yes | Absolute | 8% | 9% | 7% | 5% | 5% | 6% | 4% |
| Length | No | Relative | 9% | 9% | 11% | 10% | 10% | – | – |
| Length | No | Absolute | 9% | 6% | 7% | 6% | 4% | – | – |
| MCD1 | Yes | Relative | 5% | 3% | 2% | 4% | 3% | 3% | 2% |
| MCD1 | Yes | Absolute | 4% | 1% | 3% | 2% | 2% | 2% | 1% |
| MCD1 | No | Relative | 4% | 4% | 3% | 2% | 2% | – | – |
| MCD1 | No | Absolute | 5% | 2% | 1% | 1% | 2% | – | – |
| MCD2 | Yes | Relative | 9% | 13% | 9% | 5% | 4% | 7% | 3% |
| MCD2 | Yes | Absolute | 5% | 6% | 2% | 2% | 2% | 2% | 2% |
| MCD2 | No | Relative | 13% | 7% | 4% | 5% | 4% | – | – |
| MCD2 | No | Absolute | 4% | 4% | 2% | 2% | 1% | – | – |
| MCD3 | Yes | Relative | 12% | 12% | 6% | 5% | 7% | 2% | 3% |
| MCD3 | Yes | Absolute | 3% | 4% | 3% | 3% | 3% | 2% | 2% |
| MCD3 | No | Relative | 8% | 8% | 7% | 3% | 2% | – | – |
| MCD3 | No | Absolute | 4% | 5% | 2% | 3% | 1% | – | – |

Figure 18: Test accuracy by number of layers on SCAN splits when trained using standard, end-to-end supervision with weight sharing and relative positional encodings. Test accuracy does not increase as the number of layers increases.

