# OpenReview forum: "ALTA: Compiler-Based Analysis of Transformers"
_TMLR — Accepted by TMLR_

### Review · Reviewer_XoJj · 2024-11-21

**Summary Of Contributions:**

This paper introduces ALTA, a new programming language and compiler that maps programs to Transformer weights, extending previous work (RASP and Tracr) with support for loops without intermediate decoding steps and compilation to Universal Transformers. The authors demonstrate novel expressivity results by showing how Transformers can represent length-invariant algorithms for parity and addition, as well as a solution to the SCAN benchmark, all without requiring intermediate decoding. To analyze the gap between expressivity and learnability, they introduce tools including trace supervision for analyzing when algorithms can be learned from training data, and a theoretical framework for analyzing program minimality. Through empirical analysis, they show that Transformers trained end-to-end tend to learn simpler algorithms requiring fewer layers rather than more complex algorithms that generalize better, and demonstrate that increasing model depth does not necessarily improve compositional generalization. The authors make ALTA available as open source, enabling further research into Transformer capabilities and limitations. In general, this paper seems make a good contribution to this field and doing a very thorough experiment.

**Audience:**

Yes

**Broader Impact Concerns:**

I do not have any concerns about the broader impacts for this paper.

**Claims And Evidence:**

Yes

**Requested Changes:**

Please refer to my weakness part.

**Strengths And Weaknesses:**

Strengths:
- ALTA's sparse representation enables compilation of complex programs into reasonably-sized Transformers by leveraging the sparsity of transition rules to reduce required MLP hidden dimensions, sometimes by several orders of magnitude. This is in contrast to RASP's dense lookup table approach which can suffer from combinatorial explosion in variable combinations.

- The representation of MLP sublayer computations as sparse transition rules provides valuable insights into the generalization potential of MLP layers and Transformers as a whole, offering a new lens to understand their behavior.

- The paper constructively demonstrates how Transformers can solve the SCAN task through a shift-reduce parsing algorithm, proving that Transformers can represent systematic generalization through a finite set of transition rules and attention operations.

- The set of transition rules in ALTA programs provides a new metric for measuring Transformer complexity, enabling studies of whether Transformers have an inherent simplicity bias by comparing the number of transition rules across different programs.

- The work establishes a solid theoretical framework for analyzing minimality and learnability, effectively bridging the gap between expressivity and practical implementation.

Weaknesses:

- While the paper presents attention as a mechanism for computing binary selection matrices and aggregating information across positions, recent advances in MetaFormer [R1] architectures suggest that simpler operations like pooling could potentially replace attention. Similarly, State Space Models and Linear Transformers have demonstrated success using recurrent structures instead of traditional attention. The paper would benefit from exploring whether ALTA's framework could extend to these alternative architectures, potentially offering insights into the essential computational requirements for tasks like SCAN and parity.

- Although the paper mentions that ALTA programs can be executed in decoder-only Transformers by adding causal attention masks and outer auto-regressive decoding loops, this aspect receives limited discussion. A more detailed analysis of decoder-only implementations would be valuable, particularly given the growing prominence of decoder-only models in contemporary research.

[R1]: Yu, Weihao, et al. "Metaformer is actually what you need for vision." Proceedings of the IEEE/CVF conference on computer vision and pattern recognition. 2022.

---

> ### Author Response · Authors · 2024-12-17
>
> Thank you for your review!
>
> > Extending ALTA to study alternative architectures
>
> We certainly agree that this is an excellent direction for future work! The design and implementation of ALTA is relatively modular between components related to the self-attention vs. other sub-layers (e.g. MLPs), so we hope our work can support such an investigation.
>
> Given the current centrality of self-attention in the field, we believe that focusing on this architecture is a sufficient scope for the current paper.
>
> > Analysis of Decoder-only Transformers
>
> While we focused on analysis of encoder-only Transformers in this paper, we agree that applying ALTA to study decoder-only Transformers would be an interesting direction for future work. We mentioned in Section 2 the two modifications needed to extend ALTA to such a context (a casual attention mask, and an outer auto-regressive decoding loop), but will provide more details in the appendix to better support future work in this direction.

---

### Review · Reviewer_SgJx · 2024-11-29

**Summary Of Contributions:**

This work presents a new programming language (ALTA) and its compiler that compiles symbolic programs to Transformer weights. Based on previous work (RASP language), ALTA additionally allows loops to be represented with layer-wise Transformer weight sharing. This mechanism enables mapping programs to Universal Transformers and the implementation of length-invariant algorithms. In addition, ALTA is more efficient than previous approaches as it can compile more complex programs to reasonably sized Transformers. This is done by representing MLP layers with sparse transition rules.

Overall this work presents new theoretical and empirical insights on the representation power and learnability of Transformers.

**Audience:**

Yes

**Broader Impact Concerns:**

no concern

**Claims And Evidence:**

Yes

**Requested Changes:**

- [*critical to secure my recommendation*] Clarify the SCAN results: mention the accuracies of the ALTA compiled transformer, if it is not 100%, then rephrase all occurrences saying that you “solve” the SCAN benchmark (see weakness paragraph 3)

- [*to strengthen the work*] Accuracy numbers on Addition (and SUBLEQ if that is possible) (see weakness paragraph 2)

**Strengths And Weaknesses:**

**Strengths**

This paper is well-organized, and relatively easy to follow given the complexity of the proposed approach. The experiments on the parity problem provide some evidence of the effectiveness of the proposed framework.

This work presents a theoretical analysis on both the expressivity of Transformers, and the learnability of algorithms. Even if an algorithm can be expressed with a Transformer class, training such model and making it generalize beyond its training set can be challenging. This work shows that training on execution traces from an ALTA program can improve performance for some tasks.

This work introduces the analytical notion of *minimal* programs with respect to a training set: a minimal program will be compiled to a Transformer which uses all of its parameters fully to represent the training set. A non-minimal program will be compiled to a set of weights that can be changed without affecting the prediction power of the training set, ie: some parameters become under-specified.

**Weaknesses**

One weakness of this work, as recognized in the paper and as with similar approaches, is the relatively small variety of representable Transformers by the ALTA framework. For instance, the selection matrix of the self-attention layers can only be binary. There is limited support for numeric computations and no support for probabilistic output distributions.

Another weakness is the small amount of experimental results on different tasks. The paper shows that the ALTA framework allows to generalize to various difficulty levels of the parity problem, but does not show the same thing for the addition, SUBLEQ, and SCAN tasks. Instead, the ALTA programs are presented in respective Appendices, but it is not clear if the compiled Transformer, for each of these task achieves a strong performance across all difficulty levels. Does having an ALTA program automatically guarantee that the problem is solved (ie: 100% accuracy regardless of the input difficulty)? Some clarifications around this and the deterministic nature of the approach would greatly improve the paper.

In particular, in the case of the SCAN benchmark the paper claims multiple times that it “solves” the SCAN benchmark, however, when looking at the results in Figure 18 and Table 5, we see that the best accuracy achieved by training a model is 13%. Does the trained model use ALTA traces? it would be nice to show this experiment, otherwise the motivation for training a model is not clear. What are the accuracies of the ALTA compiled transformer? If it is 100% by design of the ALTA program, it must be clearly stated. Until this is clarified, I cannot say that *the claims made in the submission are supported by accurate, convincing, and clear evidence*.

---

> ### Author Response · Authors · 2024-12-17
>
> Thank you for your review!
>
> We respond to your concerns below, and will clarify the confusion related to the accuracy of compiled models in our revision. Thank you for raising this.
>
> > "Does having an ALTA program automatically guarantee that the problem is solved (ie: 100% accuracy regardless of the input difficulty)?"
>
> In all cases across parity, addition, SCAN, and SUBLEQ, the compiled models have 100% accuracy across all instances on which we evaluated them.
>
> The behavior of the compiled models matches the behavior of the original program when executed within the symbolic interpreter. While there is technically some degree of numerical approximation with respect to computing the attention matrices in the compiled models (with respect to the purely binary matrices assumed by the symbolic program interpreter), the approximation error can be driven towards zero by increasing the compiler hyperparameter discussed in Section 2.2. Regardless, it does not seem to affect the accuracy of the compiled models in any case that we observed.
>
> > "[critical to secure my recommendation] Clarify the SCAN results: mention the accuracies of the ALTA compiled transformer, if it is not 100%, then rephrase all occurrences saying that you “solve” the SCAN benchmark (see weakness paragraph 3)"
>
> Per the point above, the accuracy is indeed 100%. Thanks for flagging this, and we will clarify this.
>
> > "[to strengthen the work] Accuracy numbers on Addition (and SUBLEQ if that is possible)"
>
> Again, the accuracy of our compiled models is 100% on all instances. We verified correctness on these tasks using a collection of “unit tests”, i.e. a sample of representative inputs and outputs. We will clarify this.
>
> Also note that prior work (Section 4.2 Paragraph 1) has already shown that length generalization for addition is challenging when training models with end-to-end supervision.
>
> > Do the SCAN results in Table 5 use intermediate supervision?
>
> No, they do not. Having shown that a solution to SCAN is *expressible* by a Transformer, our motivation was to determine whether increasing the number of layers in a Transformer towards the number of layers used by our ALTA program would lead to improved performance when training models end-to-end (it did not). We will clarify this.
>
> > “One weakness of this work, as recognized in the paper and as with similar approaches, is the relatively small variety of representable Transformers by the ALTA framework.”
>
> We acknowledge this weakness, however we also note that ALTA is not meant as a full characterization of all capabilities of transformer models; rather its purpose is to identify conditions under which it is possible to guarantee the existence of Transformers that can express/learn compositionally-general algorithms, which is currently a topic of significant contention in the literature and even in popular discourse.

---

> > ### Comment · Reviewer_SgJx · 2024-12-18
> > **Thanks for clarifications**
> >
> > Thank you for clarifying and answering my questions.
> > I updated my ratings to Claims And Evidence: Yes & Audience: yes

---

### Review · Reviewer_rgVx · 2024-12-07

**Summary Of Contributions:**

This work propose a new programming language called ALTA and a compiler that can map ALTA programs to Transformer weights. Also, this work complements and extends this prior work, offering the ability to express loops and to compile programs to Universal Transformers, among other advantages.

**Audience:**

Yes

**Claims And Evidence:**

Yes

**Requested Changes:**

Please check the weakness.

**Strengths And Weaknesses:**

Strengths: The atuhor conducts experiments to try to support their claims.

Weakness:
* **The paper writing is not clear**
   * It seems that the task of this work is not very common. Therefore, after reading the paper several times, I still find it difficult to know what the task is.
      * What is the input and what is the output?
      * And moreover, could you please suggest what the task belongs to? Is it a sequence classification task, as the author uses an encoder-only transformer?
* The work is based on RASP. Therefore, which one is better for RASP and this work?
* **For Experiment Part**
   * How the dataset is constructed?
   * It seems that the author does not clearly present the dataset tasks. For example, what is Sum + Modulo? Could you give an example?
   * For Section 4.2, it seems that the dataset is Addition.
      * Traditionally, the addition task could be regarded as a next-token prediction task. And we usually solve it via a decoder-only Transformer.
      * Therefore, how does ALTA solve such ta ask?

As I have several above concerns, I have to ask the author to give further explanation.

---

> ### Author Response · Authors · 2024-12-17
>
> Thank you for your review!
>
> We hope our responses below will address your questions regarding the tasks and datasets used in the paper, and we will update the relevant sections in the paper accordingly to highlight these details. Thank you for raising these concerns.
>
> > “The work is based on RASP. Therefore, which one is better for RASP and this work?”
>
> ALTA and RASP are both symbolic languages designed to be compiled to Transformer weights (The Tracr compiler for RASP was separately proposed by Lindner et al., 2023). The primary advantages of the ALTA framework relative to RASP and Tracr are that it supports programs with loops, it enables compiling programs to Universal Transformers, and it can support compilation of more complex programs to reasonably sized models by leveraging sparsity in the MLP function (described in Sections 1-2 and Appendix A).
>
> These features enable new constructive expressivity results for Universal Transformers, e.g. we compile length-invariant algorithms for computing parity and addition, and we compile a program that solves the SCAN compositional generalization task. We also leverage ALTA to study the learnability of various algorithms with respect to a given training set (Sections 3-4).
>
> > Confusion about tasks and datasets
>
> In this paper we demonstrate how the ALTA framework could be used to compile programs to weights of encoder-only Universal Transformers. We present programs for parity, addition, and SCAN, which have been widely studied by prior work interested in the generalization of Transformers. We also present a program for interpreting SUBLEQ instructions, a capability previously studied by Giannou et al., 2023.
>
> We use these tasks to study both expressibility and learnability. To support our expressibility results, we do not consider a train and test split; rather we verify the correctness of the compiled models over a representative set of inputs (see task specific details below). For parity and SCAN, we also study learnability with respect to a training set. In these cases, we describe the train and test splits below.
>
> In an encoder-only Transformer, the model input is a sequence of tokens, and the model output is a sequence of tokens of equal length. We can encode the task input and task output in these sequences, using padding of the model input or output in cases where the task input and output have different lengths. We give examples for each task below, and also address your task-specific questions.
>
> ___Parity___
>
> Parity requires computing whether a given sequence is even or odd. Prior work (e.g. Hahn 2020) has studied whether Transformer encoders can compute parity. Similarly to this prior work, the parity of the input sequence is encoded in a particular output token (e.g. the last one). Example:
>
> | Token Position | 1 | 2 | 3 | 4 |
> | --- | --- | --- | --- | --- |
> | Example Output | _ | _ | _ | 0 |
> | Example Input | 1 | 0 | 1 | 0 |
>
> Where `_` denotes that the other output tokens are ignored.
>
> We study three different ALTA programs that can all be compiled to encoder-only Transformers that compute the parity of an input sequence (detailed in Appendix D.1). The `Sum + Modulo` program computes parity by using an attention head to compute the total number of ones, and then computing a mod 2 operation in the following MLP sub-layer.
>
> To study the learnability of parity, we constructed a train and test split detailed in Appendix D.1. The train set consists of examples between lengths 0 and 20, and the test set contains examples between lengths 0 and 40. We will clarify this in Section 4.1.
>
> __Addition__
>
> We also demonstrated an ALTA program that can compute multi-digit addition. While the reviewer correctly notes that addition has been studied in the context of decoder-only Transformers, it is also straightforward to compute addition in an encoder-only Transformer:
>
> | Token Position | 1 | 2 | 3 | 4 | 5 | 6 | 7 |
> | --- | --- | --- | --- | --- | --- | --- | --- |
> | Example Output | _ | _ | _ | _ | 9 | 0 | _ |
> | Example Input | START | 3 | 4 | + | 5 | 6 | END |
>
> Where `_` denotes padding tokens.
>
> We verified the correctness of our program and the compiled model for inputs of varying lengths, considering samples of pairs of integer inputs with 1-50 digits.
>
> __SUBLEQ__
>
> The SUBLEQ language is described in Giannou et al. (2023). The language executes instructions that read and write to a sequence of memory registers. We demonstrate an ALTA program that implements a SUBLEQ interpreter where each token in the encoder input and output represents a register position and encodes the value of the given register. We verify the correctness of our SUBLEQ program using unit tests, included in the attached code (`subleq_test.py`). We will add further details to Appendix D.3.
>
> (Continued in next comment)

---

> > ### Author Response · Authors · 2024-12-17
> >
> > __SCAN__
> >
> > The SCAN benchmark of compositional generalization consists of several train and test splits. It was originally proposed by Lake & Baroni (2018), and the 3 maximum compound divergence (MCD) train and test splits were added by Keysers et al. (2020). These papers provide details of how the train and test splits were generated. The most challenging splits for Transformers are the length split and the 3 MCD splits; therefore, we focused our analysis on these 4 splits.
> >
> > Similarly to addition, while the task has previously been studied using encoder-decoder or decoder-only Transformers, it is also possible to represent the inputs and outputs for SCAN in an encoder-only Transformer. As described in Section 4.4 and Appendix D.4, we also pad the input with memory tokens up to some maximum `N`. These tokens are necessary for representing variable-length data structures such as a stack, buffer, and parse tree within the Transformer activations. This also ensures there are sufficient tokens for encoding the full task output in cases where it is longer than the task input. Here is an example:
> >
> > | Token Position | 1 | 2 | 3 | 4 | 5 | … | N |
> > | --- | --- | --- | --- | --- | --- | --- | --- |
> > | Example Output | _ | JUMP | JUMP | _ | _ | … | _ |
> > | Example Input | START | jump | twice | END | PAD | … | PAD |

---

> ### Comment · Reviewer_rgVx · 2024-12-17
>
> Dear Authors,
>
> Thank you very much for your response. This really make the paper more clear. **However, I am not sure whether my following understanding is correct or not.**
>
> * Is the model a 1-layer Looped Transformer?
>    * If the answer is yes, please clearly present the loop iteration (should be similar to the transformer layer) and gradient iteration (how many loop iterations you are used to calculate for gradient). For example, in the original looped transformer paper, the loop iteration could be 12, and the gradient iteration could be 10.
> * What is the hyperparameter of RASP?
>    * To be more specific, what is the layer number and attention head? Is the attention head the same as the proposed ALTA?
>    * Please provide the training details as many as possible
> * Current understanding of the work: the main contribution of the paper is:
>    * Utilize the looped Transformer to try to replace the standard Transformer.
>    * The author validates the performance on specific datasets that may need loop iteration.
>
> The above is my current understanding. If there is anything wrong, please let me know. And if there is anything correct, please also let me know.

---

> > ### Author Response · Authors · 2024-12-18
> >
> > Thank you for your reply, and we’re glad that we were able to clarify some details.
> >
> > However, we think there is still some confusion, which we will try to further clarify. Our work is not directly related to Looped Transformers and we do not propose a method for improving Transformers. Please see inline answers below.
> >
> > > “Is the model a 1-layer Looped Transformer?”
> >
> > No, we compile ALTA programs to Universal Transformer (Dehghani et al., 2018) encoders. Universal Transformers are essentially standard Transformers with two differences: (1) layer-wise weight sharing and (2) an optional dynamic halting mechanism.
> >
> > Looped Transformers (Giannou et al., 2023) also involve layer-wise weight sharing; however not every layer uses the same weights. Instead, a sequence of N unique layers is repeated in a loop (with N=1 this is similar to a Universal Transformer). Additionally, the paper proposes a specific convention for encoding instructions, memory, and a scratchpad in the input embeddings and activations of the encoder, which we do not follow in this work. Other than the similarities in weight sharing between Looped Transformers and Universal Transformers, the only explicit connection between Looped Transformers and our paper is that both papers show how a Transformer can implement a SUBLEQ interpreter (see details in Appendix D.3).
> >
> > For our experiments in Section 4 that involve training models, we use a fixed number of layers (i.e. no dynamic halting mechanism). Therefore, training and inference are unchanged from a standard Transformer, except that parameters are shared across layers of the encoder.
> >
> > > “What is the hyperparameter of RASP? To be more specific, what is the layer number and attention head? Is the attention head the same as the proposed ALTA? Please provide the training details as many as possible.”
> >
> > It is important to note that when compiling programs written in RASP and ALTA to Transformer weights there is no *training*. Model parameters are a deterministic function of the program, specified by the Tracr compiler (for RASP programs) or our proposed ALTA compiler (for ALTA programs).
> >
> > The minimum number of attention heads, layers, and embedding and MLP dimensions required to compile the model can be determined by the compiler. Some of these properties of the compiled models for the various programs considered in this paper are noted in Section 4 and Appendix D, but we will add all of these values for every program to the revised version.
> >
> > It is not possible to directly compare these values with RASP for these tasks, because RASP does not support loops or compilation to Universal Transformer encoders.
> >
> > For the experiments in Section 4 that analyze learned models for parity and SCAN, the hyperparameters can be found in Appendices D.1 and D.4 respectively.
> >
> > (Continued in next comment)

---

> > > ### Author Response · Authors · 2024-12-18
> > >
> > > > “Current understanding of the work: the main contribution of the paper is: (1) Utilize the looped Transformer to try to replace the standard Transformer. (2) The author validates the performance on specific datasets that may need loop iteration.”
> > >
> > > We think there may be some confusion about our main contributions.
> > >
> > > Our goal is to improve our *understanding* of how Transformers can represent and learn different algorithms, not to directly propose a new *method* for improving performance (although we hope this work may inspire new methods in the future). More concretely, while our results support the notion that layer-wise weight sharing may be worth further study, we do *not* necessarily advocate for replacing the standard Transformer with a Universal or Looped version, and our results do not support this conclusion.
> > >
> > > Here is how we view our main contributions:
> > >
> > > * Our primary contribution is proposing the ALTA framework, including the language specification, the symbolic interpreter, and the compiler from programs to Transformer weights. We release the open-source code for this framework (an anonymized version is attached to the submission). (Section 2)
> > > * One application of ALTA is to show whether and how Transformers can *express* various algorithms. We demonstrate new constructive expressivity results for Universal Transformers, showing how they can express length invariant algorithms for computing parity and addition, as well as a solution to SCAN. This is interesting because prior work has shown negative results for SCAN and for length generalization for parity and addition when training Transformers end-to-end. (Section 3-4)
> > > * Finally, we show how the ALTA framework can be used to analyze the learnability of various algorithms with respect to a training set, both theoretically and empirically via trace supervision. (Sections 3-4)
> > >
> > > We believe that the ALTA framework will be a useful tool for the community (similarly to how RASP and Tracr have been) and have applications even beyond those considered in the paper. For example, Reviewer XoJj proposed extending ALTA to analyze differences in expressivity between self-attention and other architectures, including linear attention and State-Space Models.
> > >
> > > We will try to clarify this by more explicitly defining our main contributions.

---

> > > > ### Comment · Reviewer_rgVx · 2024-12-22
> > > >
> > > > Thank you for your precious response. I have change the "claim and evidence" to Yes

---

### Author Response · Authors · 2025-01-10

Thank you again to the reviewers! We have uploaded a new revision of the paper that incorporates the requested changes and suggestions:

* Added input-output details for parity to D.1
* Added input-output details for addition to D.2
* Added input-output details for SUBLEQ to D.3
* Added input-output details for SCAN to D.4
* Added evaluation details for addition to Section 2
* Added evaluation details for SUBLEQ to D.3
* Added compiled model sizes for parity to D.1
* Added compiled model sizes for addition to D.2
* Added compiled model sizes for SUBLEQ to D.3
* Added compiled model sizes for SCAN to D.4
* Added contribution summary to introduction
* Added explicit accuracy numbers for compiled models in Section 4
* Noted analysis of alternative architectures for future work in Section 5
* Clarified SCAN end-to-end results in Section 4.4
* Created Appendix A.3 with details for extending ALTA to decoder-only models and reference in Section 2

---

### Decision · Action_Editor_bb3E · 2025-01-22

**Recommendation:** Accept as is

**Comment:**

The paper advances understanding of Transformer architectures through ALTA. Compared to previous languages such as RASP, ALTA supports programs with loops, enables compiling programs to Universal Transformers and can support compilation of more complex programs by leveraging sparsity in the MLPs. Practically, the authors demonstrate how transformers can represent length-invariant algorithms and solve the SCAN benchmark, which is a non trivial result. ALTA allows to compile symbolic programs into Transformer weights with loop support and sparse transition rules, which offers an important perspective on Transformer expressivity, generalization and learnability.

**Audience:**

The framework (ALTA) for compiling symbolic programs to Transformer weights might be insightful for researchers studying model interpretability, compositional generalization, and algorithmic learning.

**Claims And Evidence:**

After the authors’ clarifications, all reviewers agreed the paper's main claims are convincingly supported. Although some reviewers initially requested more detail on experimental settings (especially SCAN and SUBLEQ), the revised text and supplementary appendices resolved those concerns.